# Podocyte GSK3 is an evolutionarily conserved critical regulator of kidney function

J.A. Hurcombe[1], P. Hartley[2], A.C. Lay[1], L. Ni[1], J.J. Bedford[3], J.P. Leader[3], S. Singh[1], A. Murphy[4], C.L. Scudamore[5], E. Marquez[1], A.F. Barrington[1], V. Pinto[1], M. Marchetti[1], L.-F. Wong[6], J. Uney[6], M.A. Saleem[1], P.W. Mathieson[1,7], S. Patel[8,9], R.J. Walker[3], J.R. Woodgett[8], S.E. Quaggin[10], G.I. Welsh[1] & R.J.M. Coward[1]

Albuminuria affects millions of people, and is an independent risk factor for kidney failure, cardiovascular morbidity and death. The key cell that prevents albuminuria is the terminally differentiated glomerular podocyte. Here we report the evolutionary importance of the enzyme Glycogen Synthase Kinase 3 (GSK3) for maintaining podocyte function in mice and the equivalent nephrocyte cell in *Drosophila*. Developmental deletion of both GSK3 isoforms (α and β) in murine podocytes causes late neonatal death associated with massive albuminuria and renal failure. Similarly, silencing GSK3 in nephrocytes is developmentally lethal for this cell. Mature genetic or pharmacological podocyte/nephrocyte GSK3 inhibition is also detrimental; producing albuminuric kidney disease in mice and nephrocyte depletion in *Drosophila*. Mechanistically, GSK3 loss causes differentiated podocytes to re-enter the cell cycle and undergo mitotic catastrophe, modulated via the Hippo pathway but independent of Wnt-β-catenin. This work clearly identifies GSK3 as a critical regulator of podocyte and hence kidney function.

[1] Bristol Renal, Dorothy Hodgkin Building, University of Bristol, Bristol BS1 3NY, UK. [2] Bournemouth University, Bournemouth BH12 5BB, UK. [3] Dunedin School of Medicine, University of Otago, Dunedin 9016, New Zealand. [4] Department of Pathology, Southern General Hospital, Glasgow G51 4TF, UK. [5] Mary Lyon Centre, MRC Harwell, Didcot, Oxford OX11 0RD, UK. [6] Translational Health Sciences, University of Bristol, Bristol BS2 8DZ, UK. [7] The University of Hong Kong, Pokfulam, Hong Kong. [8] Lunenfeld-Tanenbaum Research Institute, Sinai Health System & University of Toronto, Toronto M5G 1X5, Canada. [9] Institute of Metabolic Science, University of Cambridge, Cambridge CB2 0QQ, UK. [10] Feinberg Cardiovascular Research Institute, Northwestern University Feinberg School of Medicine, Chicago 60611 IL, USA. These authors contributed equally: J. A. Hurcombe, P. Hartley. Correspondence and requests for materials should be addressed to R.J.M.C. (email: Richard.Coward@bristol.ac.uk)

Nearly 700 million people, representing 10% of the global population, are affected by kidney disease manifesting as albuminuria or a reduction in the glomerular filtration rate (GFR)[1]. Even small increases in albuminuria (micro-albuminuria) or reductions of GFR increase the relative risk of progressing to end-stage renal failure (ESRF), having a cardiovascular event or dying[2,3]. In recent years, it has become clear that a key mammalian kidney cell in preventing albuminuria and preserving renal function is the glomerular podocyte. Podocytes are epithelial-like cells, located on the urinary side of the glomerular filtration barrier. They consist of large cell bodies with radiating major and minor processes that wrap around the filtering glomerular capillaries to form the foot processes with regularly spaced slit diaphragms, that provide the final barrier for glomerular filtration. Importantly, podocytes, similar to neurones[4], become terminally differentiated soon after birth, with minimal capacity to re-enter the cell cycle and proliferate. Mutations in over 40 human genes have been shown to cause albuminuria[5], which are expressed in the podocyte and their mutations are deleterious. Therefore, understanding important cellular processes that regulate the unique biology of this cell is of fundamental importance.

Understanding of podocyte biology has been greatly facilitated in recent years by the discovery of equivalent cell types in more tractable organisms that can be rapidly genetically manipulated to study their function. A good example of this is the *Drosophila* nephrocyte. Nephrocytes have several similarities to podocytes, including expression of many analogues of the critical mammalian slit diaphragm podocyte proteins such as nephrin (stick and stones and hibris), NEPH1 (dumbfounded), podocin (Mec 2) and CD2AP (GC31012). Nephrocytes function as endocytotic filtration cells, maintain adult haemolymph[6] and are involved in cardiac and immune homeostasis[7].

Glycogen Synthase Kinase 3 (GSK3) is a multi-functional serine/threonine protein kinase that regulates several distinct biological pathways[8]. It was initially described as a component of glycogen metabolism and was later shown to be downstream of insulin signalling. GSK3 is rapidly phosphorylated and inhibited in response to this hormone through activation of the phosphoinositide 3-kinase (PI3K) pathway, contributing to deposition of glycogen[9]. GSK3 has two major biological actions; as a scaffolding protein and a kinase enzyme to catalyse a variety of downstream targets[10].

GSK3 is evolutionarily conserved across all eukaryotic species. In *Caenorhabditis elegans* and *Drosophila melanogaster* it is encoded by a single gene[11]. In contrast, in mammals GSK3 exists as two isoforms, GSK3α and GSK3β, encoded by different genes on different chromosomes[11]. These isoforms have 85% overall structural homology with highly conserved kinase domains (97%), with the differences largely confined to the N and C terminal regions[12]. Mammalian GSK3 activity is dynamically regulated through phosphorylation of key residues. Phosphorylation at serine 21 (GSK3α) and serine 9 (GSK3β) results in reduced activity[13]. Although GSK3α and β are structurally similar they also have some distinct functions: GSK3β null mice die during late embryogenesis due to liver apoptosis and defective activation of NF-kappa B[14], together with cardiac abnormalities;[15] in contrast GSK3α null mice are viable, have a normal life span and, interestingly, exhibit enhanced insulin sensitivity when on a susceptible genetic background[16]. This suggests that, although the isoforms share structural similarity, they have differing biological functions and are not entirely redundant. Multiple cell-specific GSK3 knockout mouse models have been published that illustrate that the functions of the two mammalian GSK3 isoforms are also cell-type dependent[17–21].

Recently it has been reported that inhibiting GSK3 in the podocyte may be therapeutically beneficial for a variety of experimental renal diseases. These studies have focused on the GSK3β isoform with less consideration of the α isoform and have either used specific genetic inhibition of GSK3β exclusively in the podocyte[22] or pharmacological inhibitors such as lithium, 6-bromoindirubin-3′-oxime (BIO), and thiadiazolidinone (TDZD-8)[22–27]. The beneficial effects of these agents are postulated to be due to inhibition of GSK3β. However, there are no isoform-specific GSK3 inhibitors currently available, and those that are used inhibit both isoforms similarly. The most common GSK3 inhibitor used in clinical practice is lithium carbonate, in the treatment of bipolar disorders. Intriguingly, lithium can cause glomerulosclerosis and ESRF in some patients given this drug for prolonged periods[28,29] but the reason for this effect is unclear[30].

As GSK3 and its isoforms exhibit different roles in different cell types[17,19–21], in this study, we investigate GSK3's importance in the podocytes of mice and in the equivalent nephrocytes of *Drosophila* using genetic and pharmacological approaches. We find that GSK3 is critically important for the function of these cells both during development and in maturity. Furthermore, the evolutionary segregation of GSK3 into two isoforms (α and β) appears protective as either isoform can fully compensate for the other's loss. Mechanistically, GSK3 maintains the podocyte in its terminally differentiated form and prevents it from re-entering the cell cycle and undergoing mitotic catastrophe, modulated by Hippo pathway signals.

## Results

**Developmental genetic loss of podocyte/nephrocyte GSK3 is catastrophic**. To study the developmental importance of GSK3, podocyte-specific GSK3α, GSK3β and combined GSK3 α/β knockout (podGSK3DKO) transgenic mice were generated. This was achieved by crossing floxed GSK3α[16] and/or GSK3β mice[17] with a podocin Cre mouse[31] (Supplementary Fig. 1a). Mice were genotyped and genomic excision of GSK3α and β DNA verified (Supplementary Fig. 1b). Furthermore, GSK3 isoform protein loss was confirmed using IHC (Supplementary Fig. 1c).

All genotypes were born with normal Mendelian frequency (Supplementary Table 1) indicating that there was no pre-natal lethality. Single isoform podocyte-specific deletion of GSK3α or β, as well as deletion of three out of four of the GSK3 alleles (i.e. podCreGSK3α$^{fl/fl}$ β$^{fl/wt}$ or podCreGSK3α$^{fl/wt}$β$^{fl/fl}$) caused no discernible phenotypes, with mice surviving normally to two years of age. However, when both isoforms were simultaneously deleted (i.e. all four floxed GSK3 alleles inactivated [podCreGSK3α$^{fl/fl}$β$^{fl/fl}$ or podGSK3DKO]) the mice all died between postnatal day 10 and 16 ($p = 0.003$ [Fig. 1a]). Prior to death these mice were indistinguishable from their littermates with regard to external appearance, weight and behaviour (Supplementary Fig. 1e). However, at time of death they had pronounced renal involvement with enlarged pale kidneys (Supplementary Fig. 1f), renal failure (Fig. 1b), acidosis (Fig. 1c), and high levels of albuminuria (Fig. 1d; Supplementary Fig. 1g). Renal disease evolved rapidly after birth as evidenced by a 25-fold increase in albuminuria from P2 to P10 (Fig. 1d). Histological examination using light and transmission electron microscopy (TEM) also revealed that glomerular and renal abnormalities were initially subtle in podGSK3DKO mice, but these rapidly progressed over the first 10 days of life to show glomerulosclerosis, multiple tubular protein casts and major disruption of the glomerular filtration barrier on TEM (Fig. 1e). By P10 highly abnormal, cystic and vacuolated, glomeruli were evident in the podGSK3DKO mice (Fig. 1f).

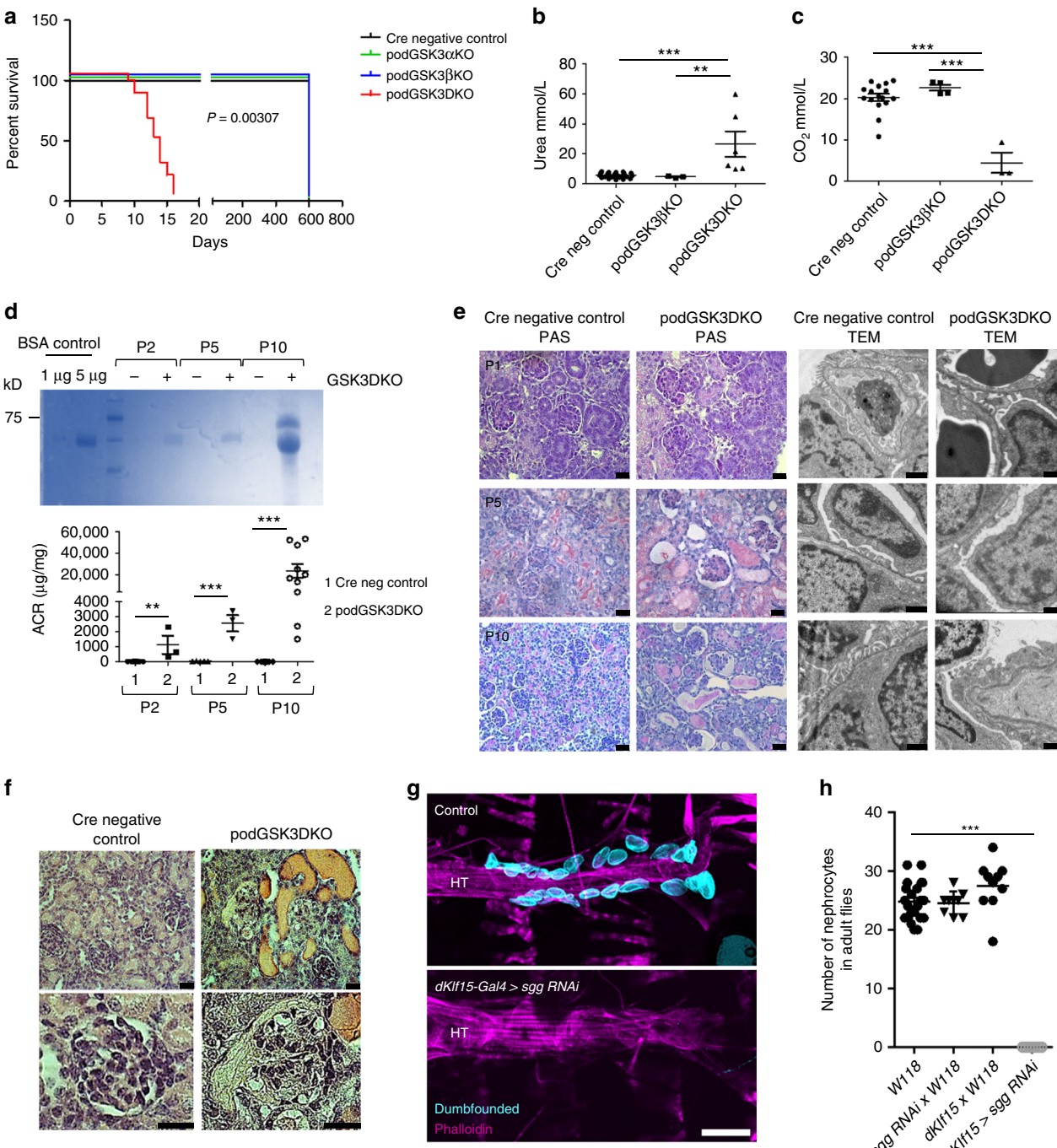

**Fig. 1** Developmental loss of podocyte/nephrocyte GSK3 is catastrophic. **a** Kaplan–Meier survival plot. Log-rank (Mantel-Cox test) $p = 0.00307$ (Cre negative controls $n = 3$; podGSK3αKO $n = 3$; podGSK3βKO $n = 3$; podGSK3DKO $n = 19$ mice). **b** podGSK3DKO mice are in renal failure with elevated urea levels at day 11 (one-way ANOVA, Tukey's post hoc test **$p < 0.01$; ***$p < 0.001$ Cre negative controls $n = 25$; podGSK3βKO $n = 3$; podGSK3DKO $n = 6$ mice). **c** podGSK3DKO mice are acidotic at day 11 with significantly lower bicarbonate levels (one-way ANOVA, Tukey's post hoc test ***$p < 0.001$ Cre negative controls $n = 16$; podGSK3βKO $n = 4$; podGSK3 DKO $n = 3$ mice). **d** Rapid progression of kidney disease in the first 11 days of life in podGSK3DKO mice. Top: a Coomassie gel showing progression of albuminuria in an illustrative knockout and control mouse (2 μl of urine loaded in each lane). See also Supplementary Fig. 1 g. Bottom: graph showing the albumin:creatinine ratio (ACR) in a population of podGSK3DKO and control mice ($t$ test **$p < 0.01$; ***$p < 0.001$ Cre negative controls P2 $n = 8$; podGSK3DKO P2 $n = 3$; Cre negative controls P5 $n = 5$; podGSK3DKO P5 $n = 3$; Cre negative control P10 $n = 10$; podGSK3DKO P10 $n = 10$ mice). **e** PAS (scale bar 25 μm) and transmission electron microscopy (scale bar $= 1$ μm) of podGSK3DKO and control mice at P1, P5 and P10. **f** H & E at P10 of podGSK3DKO and control mouse shows tubular protein casts and vacuolated glomeruli Scale bar $= 25$ μm. **g** Specific *shaggy* knockdown and control nephrocytes immunostained with dumbfounded (cyan). Scale bar $= 100$ μm. HT heart tube. **h** Nephrocyte number $n = 6$–10 flies per genotype; ANOVA, ***$p < 0.001$ for the effect of genotype. Data are presented as the mean $+/$-SEM

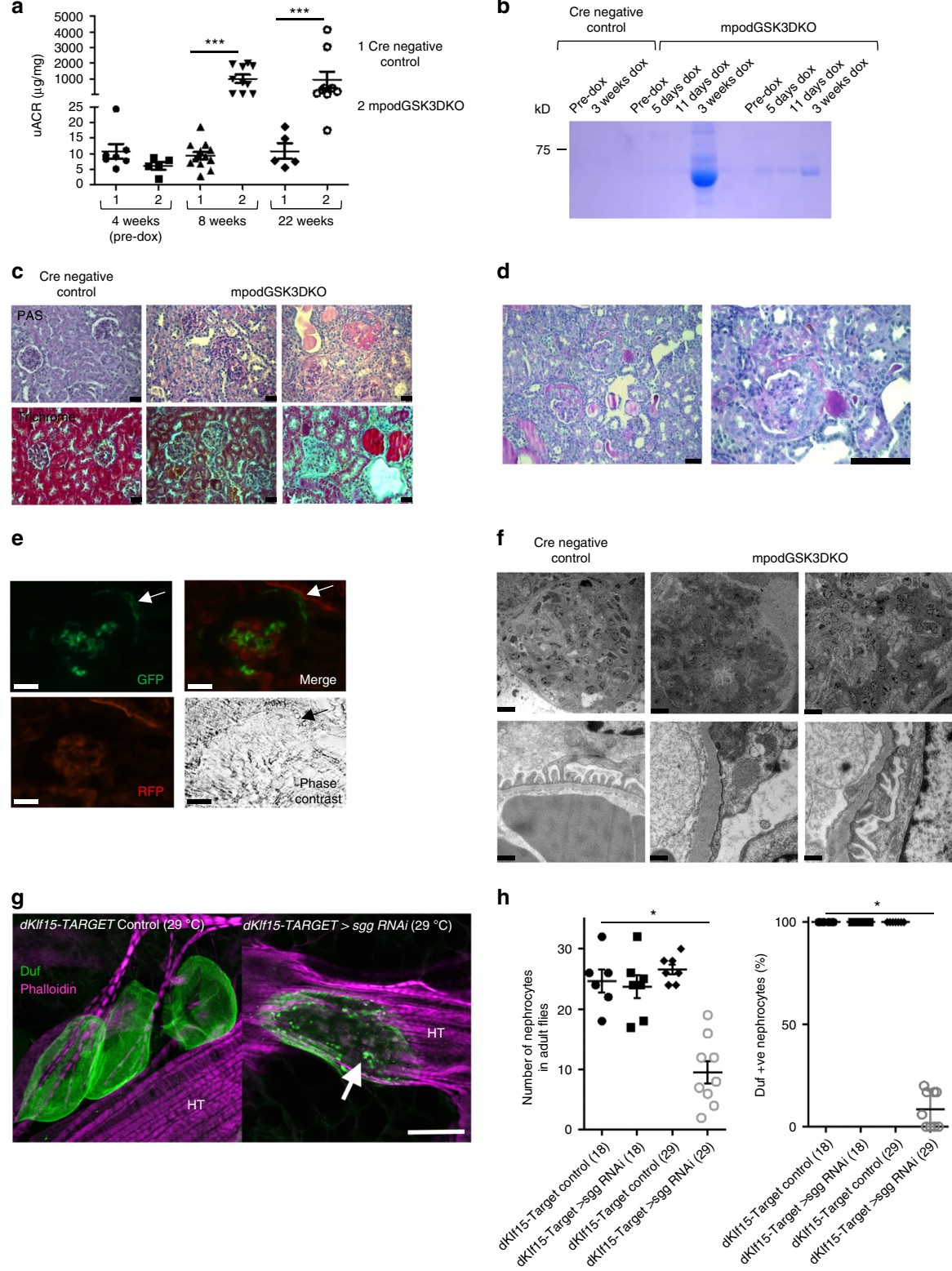

To assess whether GSK3 plays similar roles in *Drosophila* nephrocytes we used the *dKlf15-Gal4* driver to silence the GSK3 gene here, which is called *shaggy*. This resulted in a total loss of pericardial nephrocytes, indicative of a critical requirement for *shaggy* during nephrocyte development (Fig. 1g, h). Interestingly, complete absence of nephrocytes was not overtly detrimental to *Drosophila*.

**Mature genetic loss of podocyte/nephrocyte GSK3 is detrimental**. After establishing the developmental importance of GSK3 in the podocyte/nephrocyte we evaluated the role of GSK3 in mature cells. Podocin RtTA -tet-o-Cre GSK3α$^{fl/fl}$β$^{fl/fl}$ (mpodGSK3DKO) mice were generated and given doxycycline from 4 weeks of age to specifically delete GSK3 in their podocytes after glomerular development was complete (Supplementary

**Fig. 2** Mature loss of podocyte/nephrocyte GSK3 is detrimental. **a** Significant albuminuria in mpodGSK3DKO mice. Population characteristics shown. Mann-Whitney Analysis of groups ***$p < 0.001$ ($n = 9$–10). **b** Representative Coomassie staining of urine from mice. 2 μl urine loaded. **c** Spectrum of renal involvement in model. PAS and trichrome staining. Left panels = control Middle panel = least albuminuric mouse. Right = most albuminuric mouse with global glomerulosclerosis. This mouse was in renal failure with severe hypertension. Scale bar = 25 μm. See also Supplementary Fig. 3. **d** Representative picture of glomerular crescents in mpodGSK3DKO mice. Scale bar = 50 μm. **e** Lineage tagging of mpodGSK3DKO podocytes by crossing with mT/mG reporter shows extracapillary cells are originating from podocytes. GFP and merge show green fluorescent protein (GFP) tagged podocytes in Bowman's space (arrowed). Scale bar = 25 μm. **f** Glomerular filtration barrier TEM from two 22-week mpodGSK3DKO mice. Shows podocyte foot process retraction, mesangial hypercellularity and glomerular basement membrane thickening. Scale bar top panel = 10 μm; bottom panel = 500 nm. **g** Confocal micrographs of adult *Drosophila* heart stained with phalloidin (to visualise the heart tube [HT]) and antibodies to the nephrocyte marker protein Duf. Adults were reared at 18 °C until eclosure, then transferred to 29 °C. This temperature shift permits the expression of RNAi to *shaggy* specifically in nephrocytes at the higher but not the lower temperature. At 29 °C, nephrocytes in the *sgg RNAi* flies developed an enlarged morphology and abnormal foci of Duf immunoreactive staining across the cell surface (arrow). Controls at the same temperature exhibit a wild-type morphology (three nephrocytes are shown). Scale bar = 50 μm. **h** Graphs show number of nephrocytes in adult flies and percentage that were immunopositive for anti-Duf antisera. ANOVA *$p < 0.05$; $n = 6$–8 flies for each genotype at each temperature. Data are presented as the mean ± SEM

Figs. 2a–d). These mice were compared to Cre negative and non-doxycycline treated control littermates. All but one of the mpodGSK3DKO mice developed albuminuric kidney disease (Fig. 2a, b), but none of the control mice did. This occurred rapidly in the majority of mpodGSK3DKO mice with many developing albuminuria that was detectable within one week of finishing doxycycline (Fig. 2a, b). There was variability in the severity of their renal disease (Fig. 2a; Supplementary Fig. 3a), with some mpodGSK3DKO mice only mildly albuminuric at 22 weeks of age, but in all there was evidence of mesangial hypercellularity and glomerular fibrosis on trichrome staining (Fig. 2c, Supplementary Fig. 3a). In contrast ~30% of animals had raised serum creatinine relative to controls (Supplementary Fig. 3b), associated with global glomerulosclerosis and widespread interstitial fibrosis throughout the kidney (Fig. 2c; Supplementary Fig. 3a). One mouse was found to have severe hypertension (204 mmHg (SEM ± 2.67) / 158 mmHg (SEM ± 2.37) c/w systolic 115 (SEM ± 3.34) and diastolic 85 (SEM ± 2.46) in controls). mpodGSK3DKO mice had fewer podocytes and reduced nephrin expression compared to littermate controls (Supplementary Figs. 3c, d). Fascinatingly, approximately 20% of mpodGSK3DKO mice developed a florid crescentic glomerulopathy (Fig. 2d). To elucidate the cellular origin of the infiltrating crescentic cells we crossed the mpodGSK3DKO mouse with the mT/mG fluorescent reporter mouse to lineage tag the podocytes. This revealed extracapillary cells were of podocyte origin (Fig. 2e).

High-powered TEM demonstrated foot process effacement in mpodGSK3DKO podocytes together with basement membrane thickening in some mice. Low power TEM images confirmed our observations of mesangial hypercellularity and enlarged glomeruli. However, glomerular endothelial appearance and fenestrations were unchanged (Fig. 2f).

We also temporally deleted GSK3 (*shaggy*) from the mature nephrocyte in adult *Drosophila* using the *dkfl15-TARGET* system. This caused cellular dysfunction with reduced nephrocyte number and a significant loss and mislocalisation of the NEPH1 homolog dumfounded expression on the cells (Fig. 2g, h).

**Pharmacological suppression of podocyte/nephrocyte GSK3 is detrimental**. Complete genetic deletion of GSK3α and β in the podocyte was highly detrimental. However, of more clinical relevance is the pharmacological inhibition of GSK3 as there are multiple pharmacological compounds that can do this, and there is great interest in their potential use in a variety of important diseases including diabetes, cancer and dementia[32,33]. The most common agent of use in clinical practice is lithium, which is used for bipolar conditions. It is unknown how lithium precisely inhibits GSK3 activity but it is proposed that it has a dual inhibitory action by interfering with a secondary magnesium binding

site as well as increasing inhibitory phosphorylation of both serine 21 for GSK3α, and serine 9 for GSK3β[10]. We initially confirmed that lithium caused phosphorylation of these inhibitory sites in conditionally immortalised human and murine podocytes (Supplementary Figs. 4a–d). To assess the functional significance of prolonged podocyte GSK3 inhibition we then studied Wistar rats that were chronically treated with lithium for 6-months. This caused inhibitory phosphorylation of the GSK3 isoforms, together with stabilization of β-catenin specifically in their podocytes within their glomeruli (Fig. 3a). β-catenin activation occurs when GSK3 is sufficiently inhibited and thus unable to regulate the phosphorylation-dependent degradation of β-catenin via its ubiquitinylation and destruction by the 26S proteasome[21]. Lithium-treated rats developed significant proteinuria (Fig. 3b) and glomerulosclerosis (Fig. 3c, d) in comparison to their age and sex matched controls.

We then examined the direct impact of lithium on nephrocyte function in *Drosophila* using an ex vivo culture system involving semi-intact fly preparations. Within 48 h of lithium treatment dose-dependent nephrocyte hypertrophy was observed (Fig. 3e), coupled with mislocalisation of dumbfounded from uniformly short linear arrays, consistent with previous reports[34], and to an accumulation of bright, punctate Duf foci across the nephrocyte surface (Fig. 3f). Furthermore, lithium caused severe functional abnormalities to the nephrocytes' endocytic function. At lower doses (1 mM), there was an increase in the Dextran signal within nephrocytes (Fig. 3g). However, when the cells were exposed to 20 mM lithium for 48 h it resulted in major issues with their endocytic capabilities with the formation of large lacunae and uptake of large Wheat Germ Agglutinin (WGA) (Fig. 3h). In similar experiments, flies provided with food containing 5 mM lithium for 1 week showed nephrocyte loss, reduced Duf staining and increased large protein WGA accumulation (Fig. 3i). These phenotypes are consistent with progressive loss of diaphragm integrity and comparable to foot process effacement seen in injured mammalian podocytes (Supplementary Fig. 4e). Finally, we examined a kidney biopsy from a patient who had been on long-term lithium therapy and found that they also had inhibitory GSK3 phosphorylation in their podocytes compared to control human kidney samples (Fig. 3j).

**Podocyte GSK3 loss activates Wnt-β-catenin but this is not responsible for pathology**. Genetic deletion of GSK3αβ in the podocytes of podGSK3DKO mice caused substantive activation and nuclear translocation of β-catenin in this cell type. Interestingly β-catenin was also activated throughout the kidney in the tubular compartment (Fig. 4a, b; Supplementary Fig. 5a), which we subsequently identified was probably a consequence of massive albuminuria (Supplementary Fig. 5b). Similarly,

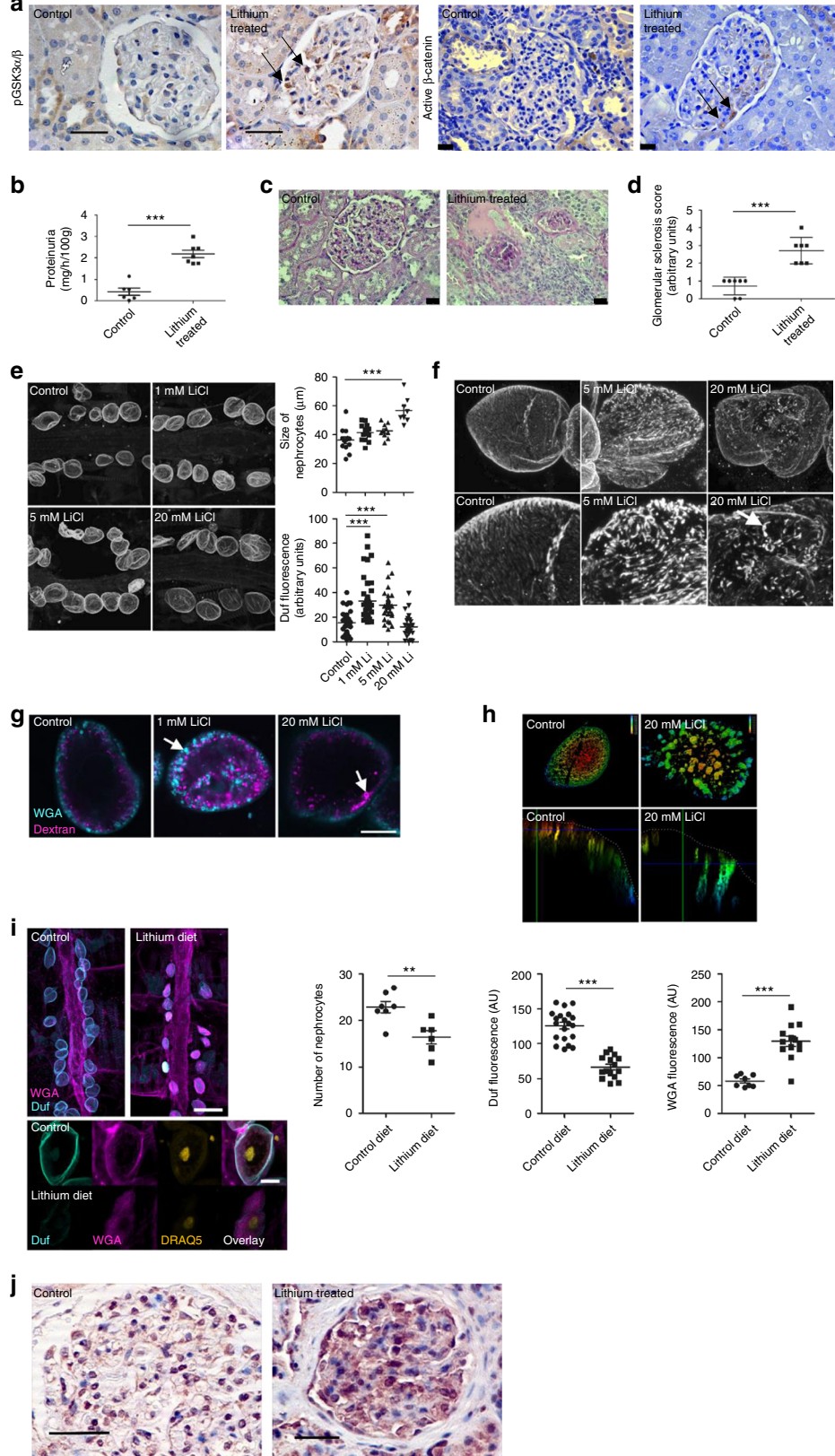

pharmacological inhibition of GSK3 also activated β-catenin in the podocyte (Fig. 3a).

As β-catenin activation and transcription of its target genes have been shown to be detrimental in other cell types when GSK3 is lost[35,36], we hypothesized that β-catenin activation was the

critical factor causing kidney damage in this model. We therefore generated transgenic mice models in which GSK3α, GSK3β and β-catenin were all contemporaneously deleted specifically from the podocyte both developmentally and in maturity. We confirmed that β-catenin was knocked down in these models

**Fig. 3** Pharmacological suppression of podocyte/nephrocyte GSK3. **a** Wistar rats fed lithium for 6-months caused inhibitory phosphorylation of GSK3α/β (left panel) and nuclear activation of β-catenin within their podocytes (right panel) (arrowed). Scale bar = 25 μm. **b** Lithium-treated rats developed significant proteinuria. t test ***p < 0.001, control n = 6; lithium treated n = 7. **c, d** Lithium-treated rats had evidence of glomerulosclerosis. Representative glomerular pictures shown, scale bar = 25 μm (**c**) together with blinded scoring of glomerulosclerosis index (**d**) in the 6-month treated Lithium and control group (n = 6–7). Unpaired two-tailed t-test*** p < 0.001. **e** At 24 h there is a dose response increase in nephrocyte size with increased Duf expression. ANOVA, ***p < 0.001, n = 6 individual flies per dose. **f** Altered cellular location of Dumbfounded in the nephrocyte after 24 h in a dose-dependent manner. Z-projected confocal stacks of controls show short linear arrays of Duf staining, presumed to be openings of individual slit diaphragms (arrowed). Cells treated with LiCl show disrupted Duf staining. **g** LiCl treatment affects endocytic Dextran uptake of dextran after 48 h. View through midpoint of cell. 1 mM demonstrates increased dextran signal uptake. At 20 mM severe disruption of dextran uptake. Cyan = dextran. Magenta = Wheat Germ Agglutinin (WGA). Scale bar = 10 μm. **h** The depth coloured images show nephrocytes stained with WGA after incubation for 48 h with or without 20 mM lithium. The uppermost region of the cell is red, the lower most is blue. In the control WGA (green) is located at the cell surface in shallow lacunae, whereas in the lithium treated cell the WGA associates with wider, deeper lacunae. Lower panels show a transection through the cell with a lacuna highlighted by an arrow. **i** Prolonged exposure of nephrocytes to lithium causes nephrocyte loss, reduced Duf expression and increased WGA nephrocyte accumulation (t test ***p < 0.001; n = 6–7 flies after 1-week exposure to 5 mM of lithium) DRAQ 5 was used to visualise nuclei. Scale bar top panel = 100 μm; bottom panel = 10 μm. **j** Immunohistochemistry showing increased inhibitory phosphorylation of GSK3αβ in the glomerulus from a patient on long-term lithium therapy when compared with a control biopsy. Scale bar = 50 μm. Data are presented as the mean ± SEM

using polymerase chain reaction (PCR), immunohistochemistry (IHC) and immunofluorescence (IF) (Supplementary Figs. 5c–e). However, a loss of podocyte β-catenin did not improve survival or kidney involvement in the developmental model (Fig. 4c–e) or renal damage in the mature model (Fig. 4f).

To support these findings, and given the contribution of Wnt/β-catenin along with Notch and Hedgehog pathways to the detrimental phenotype observed in other cells lacking GSK3[35] we also used pharmacological inhibitors of these pathways in a genetic GSK3αβ podocyte knockout cell line we developed (described in detail in next section). We inhibited Wnt signalling with iCRT3, NOTCH with DAPT and Hedgehog with GANT 61 but none of these improved cell survival (Fig. 4g–i; Fig. 8c). Interestingly, and unexpectedly, inhibiting Hedgehog signalling significantly increased cell death in this model (Fig. 4i; Fig. 8c).

**Podocyte GSK3 loss causes mitotic catastrophe.** To elucidate the mechanisms causing cellular damage when GSK3 was lost we generated a conditionally immortalised temperature-sensitive SV40 antigen podocyte cell line from a GSK3α[fl/fl]β[fl/fl] mouse. This allowed us to initially culture healthy podocytes and then temporally delete GSK3 using lentiviral delivered Cre recombinase to make a GSK3α/β knockout cell line (cipodGSK3DKO) (Supplementary Figs. 6a–c). Three days after Cre transduction there was ~85% reduction of GSK3α and a ~70% reduction of GSK3β (Fig. 5a; Supplementary Fig. 6d) but the podocytes appeared healthy without any detectable cell loss. Therefore, we elected to interrogate this time point to identify early GSK3 driven mechanistic processes. More prolonged loss of GSK3, for 7 days, caused podocytes to be lost and microscopically appear unhealthy (Fig. 5b; Supplementary Fig. 6e).

Non-biased Tandom Mass tagged (TMT) LC-MS/MS proteomic analysis of the cells 3 days after Cre transduction identified 486 proteins that were up-regulated more than 30% in cipodGSK3DKO cells compared to their controls (non-Cre treated GSK3α[fl/fl]β[fl/fl] podocytes or lentiviral Cre expressing wild-type podocytes) at a statistical significance p < 0.01. STRING (search tool for the retrieval of interacting genes/proteins) was used to visualise the protein–protein interactions and biological processes enriched in this data set. It revealed enrichment of multiple proteins involved in the cell cycle, including many involved in mitosis and mitotic spindle formation (Fig. 5c; Supplementary Fig. 7; Supplementary Table 3). To explore these findings further we performed fixed (Fig. 5d), and live (Supplementary Movie 1 and 2), phase contrast light microscopy of the cipodGSK3DKO cells. These confirmed that the cells were re-entering the cell cycle and attempting to divide as shown by significantly increased numbers of bi-nucleate and multi-nucleate cells in comparison to controls. However, many podocytes were unable to complete cytokinesis and appeared to be undergoing apoptosis and dying (Supplementary Movie 1 and 2). Detailed analysis of ciGSK3DKO cell numbers using an IN-Cell analyser confirmed these observations (Fig. 5b). The presence of multiple nuclei suggested that mitosis was occurring in ciGSK3DKO cells and western blot analysis of ciGSK3DKO cells 5 days after GSK3 knockdown showed a significant increase in phosphorylated Histone H3 (Fig. 5e and Supplementary Fig. 6f), a commonly used marker of mitosis associated with chromatin condensation[37]. Cyclin B1 is not usually detectable in healthy, mature podocytes[38] but our proteomic analysis revealed that it was one of the most up-regulated proteins in ciGSK3DKO cells (Fig. 5c and Supplementary Table 3). This was confirmed by western blotting along with increased phosphorylation of Cdk1 (Fig. 5e; Supplementary Fig. 6f). Levels of the Cyclin B1/Cdk1 complex increase during G2 of the cell cycle with dephosphorylation occurring at the G2/M transition[39]. The absence of this dephosphorylation event suggests impaired mitotic progression in ciGSK3DKO cells. cipodGSK3DKO cells then appeared to undergo apoptosis as Western blot analysis of cells at 7 days revealed increased cleavage of caspase 3 and PARP (Fig. 5f; Supplementary Fig. 6g). Collectively, these data demonstrate that cipodGSK3DKO cells were undergoing mitotic catastrophe. We also looked at our podGSK3DKO and mpodGSK3DKO models and found evidence of DNA accumulation in their podocytes with nuclear PCNA staining suggesting they were attempting to re-enter the cell cycle (Fig. 5g). Furthermore, in the mature mpodGSK3DKO mice we detected occasional mitotic figures in their differentiated podocytes (Fig. 5h).

**Hippo signalling is disrupted in GSK3 deficient podocytes.** Proteomic analysis also identified that an important protein in the Hippo-signalling pathway, Ajuba, was significantly increased by 200–300% in the ciGSK3DKO cells (Figs. 5c and 6a). This was validated by western blotting in the ciGSK3DKO cells (Fig. 6b; Supplementary Fig. 8a). Furthermore, an increased level of Ajuba was also detected in podocytes which had GSK3 pharmacologically suppressed using either lithium and a more specific GSK3 inhibitor called CHIR99201 (Fig. 6c and d; Supplementary Figs. 8b and 8c). The Hippo pathway leads to phosphorylation of YAP/TAZ proteins and prevents these proteins from translocating from the cytoplasm into the nucleus, where they bind with TEAD transcription co-activators and ultimately promote transcription of a number of genes that are associated with proliferation and apoptosis[40]. Our proteomic analysis revealed significant up-regulation of a range of YAP/TAZ TEAD targets[41] (Supplementary Table 4). Classically, when Ajuba increases it

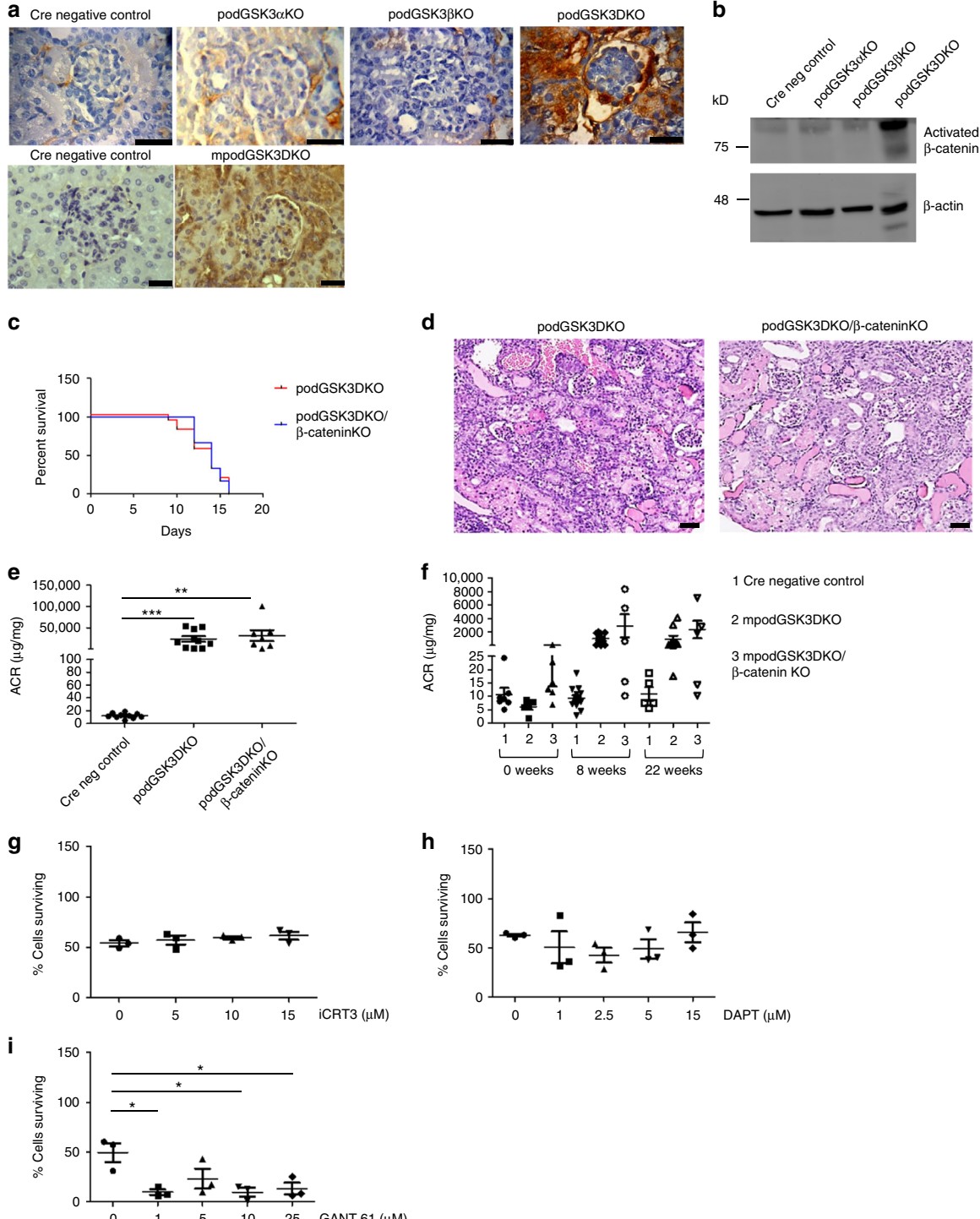

**Fig. 4** Podocyte GSK3 loss activates Wnt-β-catenin but this is not responsible for pathology. **a** β-catenin is expressed in podocytes and throughout the kidney in podGSK3DKO and mpodGSK3DKO mice but not in single isoform podocyte-specific knockout mice. Representative IHC pictures, scale bar = 25 μm. **b** Western blot of kidney lysates shows increased activated β-catenin in podGSK3DKO mice but not in control and single isoform podocyte-specific knockout mice, n = 3 experiments. See also Supplementary Fig. 5a. (**c**) No difference in survival of the podGSK3DKO/β-catenin KO compared to the podGSK3DKO mice. (Kaplan–Meier survival curve. Log-rank (Mantel-Cox test non-significant) podGSK3DKO n = 14; podGSK3DKO/β-catenin KO n = 12 mice. **d** No difference in the histological appearance of the podGSK3DKO/β-cateninKO compared to the podGSK3DKO mice at day 10 of life. Representative images of PAS staining, scale bar = 25 μm. **e** No difference in the level of albuminuria in podGSK3DKO/β-cateninKO compared to the podGSK3DKO mice. (Cre negative control n = 10; podGSK3DKO n = 10; podGSK3DKO/β-catenin KO n = 7 mice. Kruskel Wallis test not significant). **f** No difference in the level of albuminuria in mpodGSK3DKO/β-cateninKO compared to the mpodGSK3DKO mice, n = 5–7. **g** Inhibiting the Wnt pathway has no effect on cipodGSK3DKO cell survival (n = 3 experiments). **h** Inhibiting the Notch pathway has no effect on cipodGSK3DKO cell survival (n = 3 experiments). **i** Inhibiting the Hedgehog pathway in cipodGSK3DKO cells increases cell death (unpaired two-tailed t test *p < 0.05, n = 3 experiments). Data are presented as the mean ± SEM

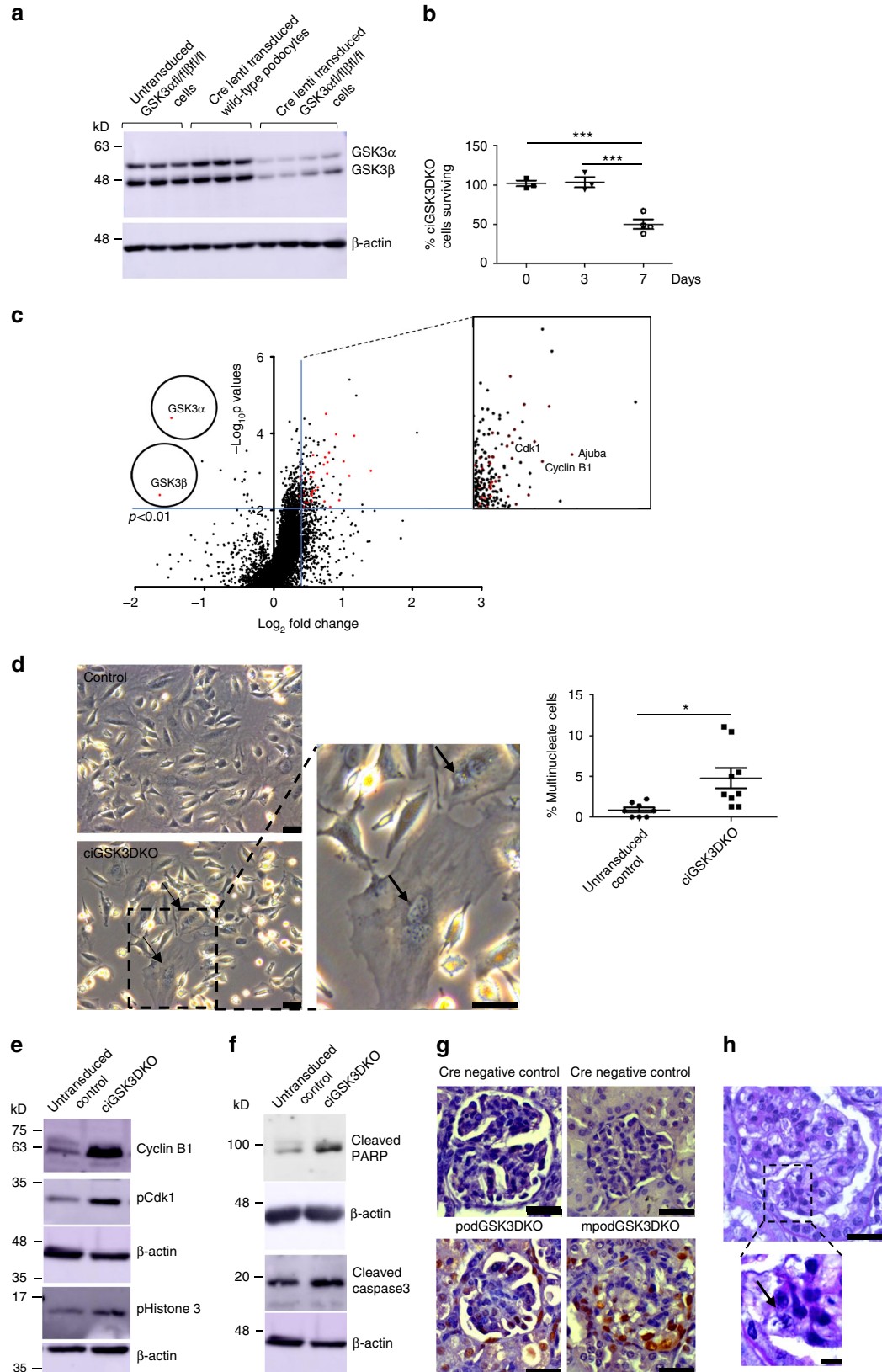

inhibits Hippo signalling by preventing the phosphorylation of YAP/TAZ thereby resulting in the nuclear translocation of these proteins[42]. In support of this mechanism we detected nuclear YAP/TAZ translocation in ciGSK3DKO cells (Fig. 6e), as well as the lithium and CHIR99201 treated podocytes (Fig. 6f and g). This was associated with induction of Cyclin B1 in all these

models (Figs. 5e and 6c, d; Supplementary Figs. 6f and 8c). Analysis of our in vivo podGSK3DKO and mpodGSK3DKO mouse models also revealed significant up-regulation of Ajuba and nuclear YAP/TAZ translocation in their podocytes (Fig. 7a–d; Supplementary Fig. 9a). This was associated with an increased number of cells positive for Cyclin A2, a cell cycle

**Fig. 5** Podocyte GSK3 loss causes mitotic catastrophe. **a** Western blot showing robust knock down of GSK3α and GSK3β in lentiviral Cre transduced floxed podocytes ($n = 4$). Controls of Cre transduced wild-type podocytes ($n = 3$) and non-cre transduced GSK3 floxed podocytes ($n = 3$) also shown. See also Supplementary Fig. 5d. **b** Only 50% of ciGSK3DKO cells survive 7 days after Cre lenti transduction. ANOVA, Tukey's post hoc test ***$p < 0.001$, $n = 3$ experiments. **c** Volcano plot of proteomic data revealed that at day-3 following lentiviral transduction, GSK3α and GSK3β levels were significantly reduced (unpaired two-tailed $t$ test $p < 0.01$). Numerous proteins associated with cell cycle entry were up-regulated. **d** Phase contrast microscopy shows ciGSK3DKO cells have significantly more multinucleate cells than controls at day 5, unpaired two-tailed $t$ test *$p < 0.05$. Three fields of view per group, $n = 3$ experiments. **e** Representative western blots of cipodGSK3DKO at day 5 reveals increased expression of Cyclin B1 and phosphorylation of Cdk1 and Histone 3 when compared with control cells, $n = 3$ experiments. See also Supplementary Fig. 6f. **f** Representative western blots of cipodGSK3DKO and control cells at day 7 reveals apoptosis in knockout cells. Increased levels of cleaved Caspase 3 and cleaved PARP shown, $n = 4$–5 experiments. See also Supplementary Fig. 6g. **g** Representative immunohistochemistry showing increased PCNA staining in glomerulus of mpodGSK3DKO mice. Scale bar $= 25 \mu m$. **h** Histology showing a mitotic figure an mpodGSK3DKO mouse podocyte. Scale bar top panel $= 25 \mu m$; bottom panel $= 10 \mu m$. Data are presented as the mean ± SEM

protein and a recognised target of YAP/TAZ-TEAD[40] (Fig. 7e). We also detected an increase of another YAP/TAZ TEAD target, c-myc[43] in both podGSK3DKO and mpodGSK3DKO mice (Fig. 7f and Supplementary Figs. 9c, d). Finally, we examined the glomeruli of our lithium treated rats and human biopsy. These both showed increased podocyte Ajuba staining (Fig.7g).

Nuclear translocation of YAP/TAZ also occurred in the podGSK3DKO/β-cateninKO and mpodGSK3DKO/β-cateninKO mice (Supplementary Fig. 9b) indicating that this process is independent of Wnt- β-catenin signalling.

We then investigated if inhibiting YAP/TAZ activity in the nuclei of GSK3α/β knockdown podocytes was beneficial. Treating ciGSK3DKO podocytes with the YAP/TAZ-TEAD inhibitor verteporfin, improved cell survival by approximately 50% demonstrating that at least some of the adverse effects of podocyte GSK3 loss are mediated through the Hippo pathway (Fig. 8a, c; Supplementary Fig. 10a). Western blot analysis showed that verteporfin was also able to attenuate Cyclin B1 accumulation in ciGSK3DKO cells (Fig. 8b; Supplementary Fig. 10b). This was also the case in lithium and CHIR99201 treated podocytes (Fig. 8d, e; Supplementary Figs 10c,10d).

Finally, we assessed the role of Hippo signalling in the nephrocytes of *Drosophila*. Verteporfin protected inducible genetic *shaggy* knockout nephrocytes from the loss and mis-localisation of dumfounded (Fig. 9a–c) and also lithium induced nephrocyte damage (Fig. 9d, e). Collectively this data supports a key role of dysregulated Hippo signalling when podocyte or nephrocyte GSK3 activity is lost (Fig. 9f).

## Discussion

Here, we have comprehensively assessed the evolutionary importance of GSK3 in podocytes and nephrocytes, which are key cells that control excretory function in mammals and *Drosophila*. Our findings show that GSK3 is critically important in these cells both developmentally and in maturity, as sufficient loss of this enzyme results in severe phenotypes in both distinct species.

We initially assessed the developmental importance of GSK3 using a genetic approach to study the mammalian podocyte in transgenic mice. This revealed a high level of compensation between the GSK3 isoforms in the podocyte. Indeed, possessing a single allele of GSK3α or β was sufficient to maintain normal function of this cell. In contrast, simultaneous loss of both isoforms caused an extreme phenotype with all mice dying in renal failure in the neonatal period. This severe phenotype was mimicked in our *Drosophila* model, where nephrocyte-specific silencing of *shaggy* resulted in the complete loss of this cell. This shows firstly that GSK3's action in these excretory cell types is fundamentally important and has likely been preserved throughout evolution. Secondly, the evolutionary duplication of GSK3 into two mammalian isoforms is beneficial for the podocyte

as it allows a high level of compensation within this system should a single isoform be lost. This contrasts to some other cell types where the β isoform appears to be the predominant mammalian isoform that maintains normal homeostatic cellular function[44–46].

We then investigated GSK3's importance in maturity. We found that mpodGSK3DKO mice developed a spectrum of kidney disease, ranging from albuminuric mesangial hyper-cellularity to glomerulosclerosis, severe hypertension and renal failure. In some mice a proliferative crescentic glomerulopathy ensued, suggesting that the terminally differentiated podocyte cells had re-entered the cell cycle. We suspect the variety of kidney disease we observed was due to variable degrees of efficiency of excision of GSK3 in this model, and because the mice were on mixed genetic backgrounds. The crescentic glomerulopathy in 20% of mice is interesting as the capacity of the mature podocyte to re-enter the cell cycle and proliferate is widely debated and controversial, but our studies support previous work suggesting that terminally differentiated podocytes can re-enter the cell cycle, divide and form crescents in inflammatory glomerulonephritis[47] and in HIV associated nephropathy[48]. Conceivably, GSK3αβ loss in the podocyte could be a common final pathway in a variety of cres-centic conditions and this is currently under investigation. Our mature *Drosophila* studies confirmed that GSK3 is also important for maintaining nephrocyte function after development in this species, showing this is also evolutionarily conserved.

After evaluating the key role of podocyte/nephrocyte GSK3 in our genetic models we investigated whether in vivo pharmaco-logical inhibition of GSK3 was also detrimental to excretory function, which we found to be the case both in rodents and *Drosophila*. We did this for three reasons: firstly because recent experimental work has suggested that pharmacologically inhi-biting podocyte GSK3 activity could be beneficial in treating a variety of renal diseases, predominantly through suppression of the β isoform;[22,25,26] secondly because there is great therapeutic interest in pharmacologically inhibiting GSK3 in a number of important disease processes including type 2-diabetes[49], bipolar disorders[21], Alzheimer's disease[21], cardiovascular disease[50] and several cancers[51], thereby making it likely that new GSK3 inhi-bitors will be trialled in patients in the future; and thirdly because some pharmacological GSK3 inhibitors are currently already widely used in clinical practice. The commonest of these is the mood-stabilizer lithium[52]. This drug is normally well tolerated, and when administered at doses that achieve therapeutic levels, only suppresses cellular GSK3 activity by approximately 25%[21]. Unfortunately some patients taking lithium develop renal side effects, the most common being diabetes insipidus, but the most severe is ESRF, which is 6–8 times more common in these patients, especially when they have taken the medication for a prolonged period of time[53–55]. It has previously been postulated that lithium-associated ESRF is due to interstitial fibrosis[56],

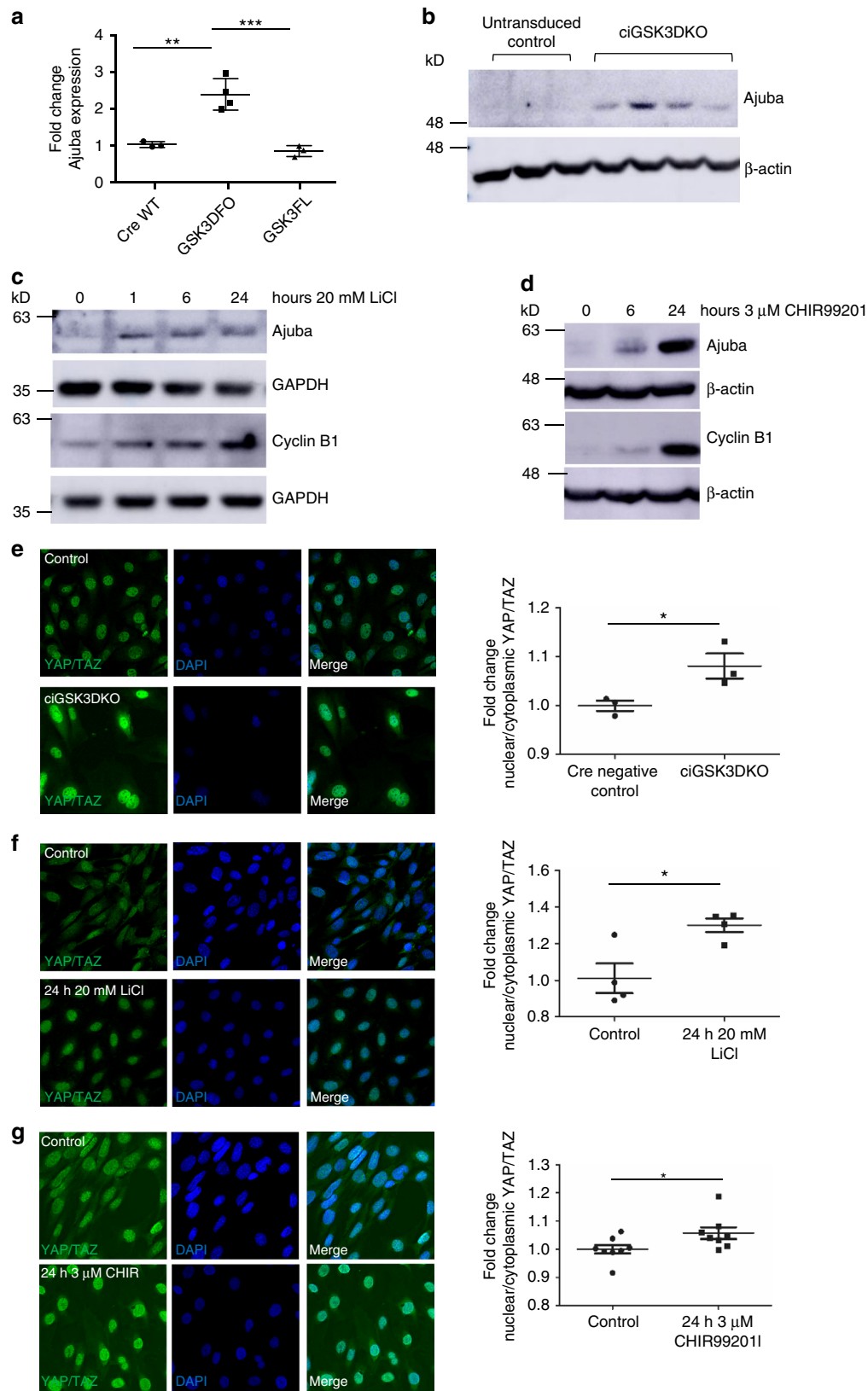

however, there are multiple reports showing that glomerulosclerosis also commonly co-exists as renal function deteriorates[28,29,57]. We suspect that in some patients receiving lithium there is excessive suppression of GSK3α/β activity in their podocytes and this leads to glomerular and kidney damage. Going

forward it will be desirable to define precisely which downstream pathways are differentially regulated by GSK3α and GSK3β in the podocyte as this could elucidate both the beneficial and detrimental pathways that are regulated by this multi-functional enzyme. Our work, and the recent studies proposing the potential

**Fig. 6** Hippo signalling is disrupted in GSK3 deficient podocytes in vitro. **a** Summary of proteomics results for Ajuba in Cre treated wild-type (CreWT $n =$ 3), Cre treated floxed GSK3α/β (GSK3KO $n = 4$) and non-Cre treated floxed GSK3α/β (GSK3FL $n = 3$) podocytes. ANOVA $p = 0.004$, Tukey post hoc analysis **$p < 0.01$ ***$p < 0.001$. **b** Representative western blot of analysis of ciGSK3DKO cells shows increased expression of Ajuba relative to controls, $n = 3–4$. See also Supplementary Fig. 7a. **c**, **d** Representative western blots of wild-type mouse podocytes incubated with 20 mM LiCl (**c**) and 3 μM CHIR99201 (**d**) showing increased expression of ajuba and Cyclin B1, $n = 3$. See also Supplementary Figs. 7b and c. **e–g** Immunofluorescence analysis showing increased nuclear YAP/TAZ staining in ciGSK3DKO cells 24 h after induction of gene knockout (**e**) $n = 3$, wild-type podocytes after 24 h 20 mM LiCl (**f**) $n = 4$ and 24 h 3 μM CHIR99201 (**g**) $n = 8$. Unpaired $t$ test *$p < 0.05$. Data are presented as the mean ± SEM

therapeutic benefit of suppressing the β isoform alone[22,25,26], also support the development of isoform-specific GSK3 inhibitors as it is possible that these could be therapeutically beneficial, without the unwanted side-effects associated with excessive suppression of both GSK3α and β. Finally, it may be prudent to monitor the levels of albuminuria in patients who have been on long-term or high-dose GSK3 inhibitors, such as lithium. Detecting albuminuria would suggest excessive GSK3 inhibition in the podocyte and may result in reduction of the drug dose or change of therapeutic approach. This could conceivably prevent the patient from developing chronic or end-stage renal failure.

Mechanistically, we initially hypothesized that the activation of β-catenin within the podocyte was the major pathway causing cellular damage when GSK3 activity was lost here, as this is the case in other cell types lacking GSK3, including neurones[35] and mammary cells[36]. Furthermore, podocyte β-catenin deletion has been shown to be protective in a variety of experimental glomerular diseases[58], suggesting that its activation is detrimental. However, we did not find that β-catenin activation was responsible for the detrimental pathology we observed. Instead we discovered that a lack of podocyte GSK3αβ caused these terminally differentiated cells to re-enter, but not exit, the cell cycle and to undergo mitotic catastrophe. We were surprised that this differed from neural progenitors lacking GSK3α/β, as these cells have several similarities to podocytes including being terminally differentiated[4]. Neural progenitors lacking GSK3α and β proliferate uncontrollably due to activation of Wnt-β-catenin, Notch, and Hedgehog pathways[35], which is not the case in podocytes. Rather, podocytes enter, but fail to exit, the cell cycle and then undergo apoptosis as a result of mitotic catastrophe in a manner similar to mature cardiomyocytes that completely lack GSK3[59]. We suspect that the necessity of GSK3 in controlling cell cycling in the podocyte explains why our podGSK3DKO mice were initially unaffected at birth but then developed severe kidney disease and subsequently died after 10–16 days. This indicates that GSK3 activity is important in the later stages of podocyte development, facilitating transition from a proliferating to a functional, terminally differentiated cell. We do not know of any other gene that has been developmentally knocked out in the podocyte that causes a similar phenotype.

Sufficient genetic or pharmacological suppression of GSK3 in the podocyte caused the Hippo pathway to be switched off, preventing the phosphorylation of YAP/TAZ, resulting in its subsequent nuclear translocation. Importantly, inhibiting the effects of nuclear YAP/TAZ translocation with verteporfin improved podocyte cell survival in ciGSK3DKO cells, showing that this is an important pathway modulated by this enzyme. This was also the case in genetic or pharmacologically GSK3-suppressed nephrocytes showing again that this is an evolutionarily conserved pathway in these cell types.

Interestingly, our results show that nuclear YAP/TAZ translocation caused by GSK3 loss in podocytes is independent of Wnt/β-catenin signaling in contrast to observations in other cell types[60]. We note that inhibiting nuclear YAP/TAZ actions only rescued survival in ciGSK3DKO podocytes by 50% and that inhibiting Notch, Wnt-β catenin, or Hedgehog did not enhance the survival of the cells. It will therefore be interesting to define the other critical pathways regulated by GSK3 in the podocyte.

In conclusion, this study has shown that GSK3 is a critical regulator of cellular function in podocytes and nephrocytes. Mechanistically it is important for maintaining the differentiated phenotype of these cells and preventing them from re-entering the cell cycle resulting in apoptosis through mitotic catastrophe. If treating patients with GSK3 inhibitors, care needs to be taken to ensure adequate GSK3 function remains in the podocyte in order to protect against the development of severe kidney disease.

## Methods

**Rodent models**. Mice in which exon two of GSK3α[16] or GSK3β[17] has been flanked by loxP sites were crossed with podocin-Cre mice[31] to generate podocyte-specific GSK3α and/or GSK3β knockout animals from embryonic day 12 (podGSK3DKO mice) (Supplementary Fig. 1a for breeding scheme). Cre negative mice, or mice with 2 or 3 of the 4 GSK3 alleles inactivated, served as controls. For the maturity model podocin rtTA and tet-o-Cre[61] mice were crossed with GSK3α/β floxed mice to generate doxycycline inducible podocyte specific GSK3α/β knockout mice (mpodGSK3DKO mice) (Supplementary Fig. 2a for breeding scheme). To induce GSK3 deletion, 4-week old mice were given doxycycline (Sigma) via their drinking water (2 mg/ml doxycycline in 5% sucrose) for 3 weeks. Controls were podocin rtTA or tet-o-Cre negative mice together with podocin rtTA and tet-o-cre positive but without floxed GSK3 genes who also received doxycycline. We confirmed the cellular specificity of the Cre drivers by crossing the podocin-Cre mouse and the pod rtTA tet-o-cre mouse with a membrane-targeted dimer Tomato/ Cre mediated membrane-targeted Green Fluorescent Protein reporter mouse (mT/mG)[62] (Supplementary Figs. 1d and 2b).

For the triple GSK3α /GSK3β /β-catenin podocyte specific transgenic mice we also crossed in the floxed β-catenin mouse floxed *Ctnnb1*gene encoding β-catenin[63] (The Jackson Laboratory).

All mice were on a mixed genetic background with contributions from 129/SV, FVB, DBA2J and C57BL/6. Both sexes were studied, and no phenotypic differences observed.

Transgenic mouse work was carried out in accordance with the University of Bristol's institutional guidelines and procedures approved by the United Kingdom (UK) Home Office in accordance with UK legislation.

For the rat experiments seven male Wistar rats were given lithium at an initial dose of 40 mmol lithium/kg dry food for 7 days and then 60 mmol lithium/kg dry food until 6 months of age. Rats given lithium had access to a salt block to maintain sodium balance and prevent lithium intoxication. This resulted in the rats having lithium levels similar to those found in humans of 0.3 to 1.3 mmol/l. These rats were compared to 6 age-matched control male Wistar rats given standard rodent diet (Speciality Foods, Perth, Australia) for 6 months. This study had ethical approval from the University of Otago Animals Ethics committee under New Zealand National Animal Welfare guidelines.

**Drosophila husbandry**. The Oregon R wild type *Drosophila* stock, *tub-Gal80^ts^* and *shaggy* RNAi lines were ordered from Bloomington. *dKlf15-Gal4* flies and *dKlf15-TARGET* flies were used as previously described[64]. For temporal *shaggy* knockout, adults were reared at 18 °C until eclosure, then transferred to 29 °C. This temperature shift permits the expression of RNAi to *shaggy* specifically in nephrocytes at the higher but not the lower temperature. Driver and RNAi over-expression combinations were generated by standard crosses. Flies were maintained under standard laboratory conditions (12 h: 12 h light dark, @ 25 °C, with ad lib access to food).

**Cell lines**. Conditionally immortilsed podocyte cell lines were cultured in RPMI1640 supplemented with 10% Fetal Bovine Serum and 1% penicillin/streptomycin. Cells were initially grown at 33 °C to allow proliferation before switching to 37 °C for 10–14 days to allow cells to differentiate. Cell lines were authenticated by western blotting for typical podocyte marker proteins.

**End point PCR**. Genomic DNA was isolated from mouse tissues using a blood and tissue DNA extraction kit (Qiagen). 500–1000 ng DNA was used in end point PCR reactions using hotmaster taq polymerase (5 Prime).

**Urine and blood analyses**. Gross albuminuria was detected using Coomassie staining of 2 μl of urine run on an SDS-PAGE gel and comparing to bovine specific

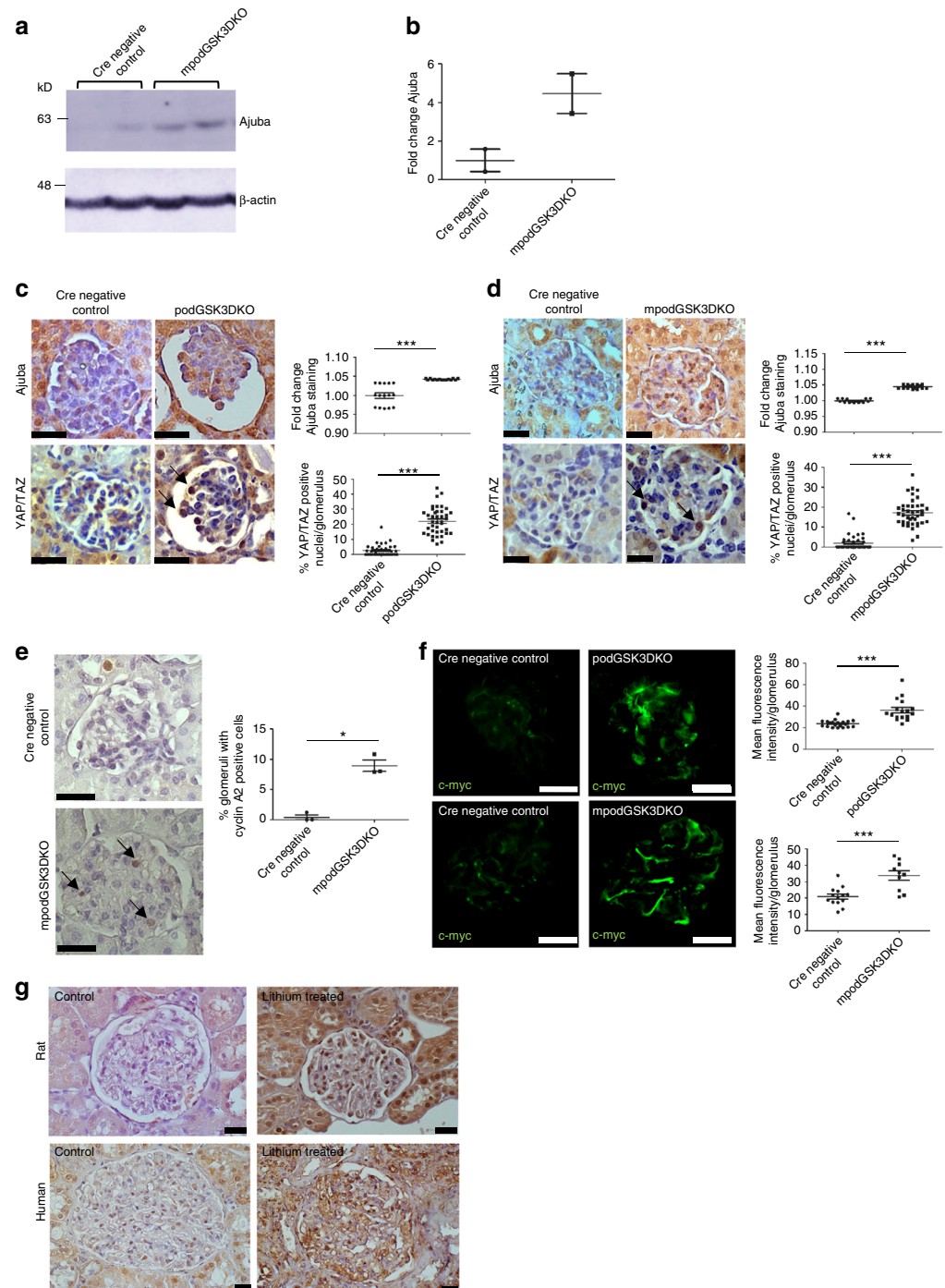

**Fig. 7** Hippo signalling is disrupted in GSK3 deficient podocytes in vivo. **a**, **b** Western blot (**a**) and quantification (**b**) of glomeruli isolated from mpodGSK3DKO and littermate control mice showing increased expression of Ajuba, $n = 2$ mice per group. **c**, **d** Immunohistochemistry showing increased Ajuba (top panels) and YAP/TAZ nuclear translocation (arrowed, bottom panels) in the podocytes of podGSK3DKO (**c**) and mpodGSK3DKO (**d**) mice. Representative immunohistochemistry and quantification shown (>5 glomeruli per mouse analysed, 3 mice per group, ANOVA ***$p < 0.001$. Scale bar = 25 μm). **e** Immunohistochemistry showing an increase in the number of cells positive for the YAP/TAZ TEAD target Cyclin A2. Three mice from each group analysed, $t$ test *$p < 0.5$. **f** Immunofluorescence staining showing increased expression of c-myc in podGSK3DKO and mpodGSK3DKO mice. Three mice analysed per group, $t$ test ***$p < 0.001$. Scale bar = 50 μm. **g** Representative immunohistochemistry using an anti-Ajuba antibody in a glomerulus from a rat given high dose lithium for 6 months and in a biopsy from a patient on long-term lithium therapy. Ajuba expression is increased with lithium treatment relative to control tissue. Scale bar = 25 μm. Data are presented as the mean ± SEM

albumin controls of 1 μg and 5 μg run on the same gel. Quantitative analysis was also performed using a mouse-specific albumin ELISA (Bethyl) and creatinine companion kit (Exocell), following the manufacturers methodology. Serum creatinine, urea and $CO_2$ was measured using the Roche Cobas system with reagents and protocols supplied by the manufacturer.

**Histology and electron microscopy**. Tissues were fixed in 10% buffered neutral formalin, further processed and paraffin embedded. 3 μm sections were stained using Periodic acid Schiff (PAS) and Masson's trichrome staining kits (Sigma) or by haematoxylin and eosin (H&E) using standard techniques. Tissues were imaged using a Leica DM2000 microscope and micrographs taken with Leica Application

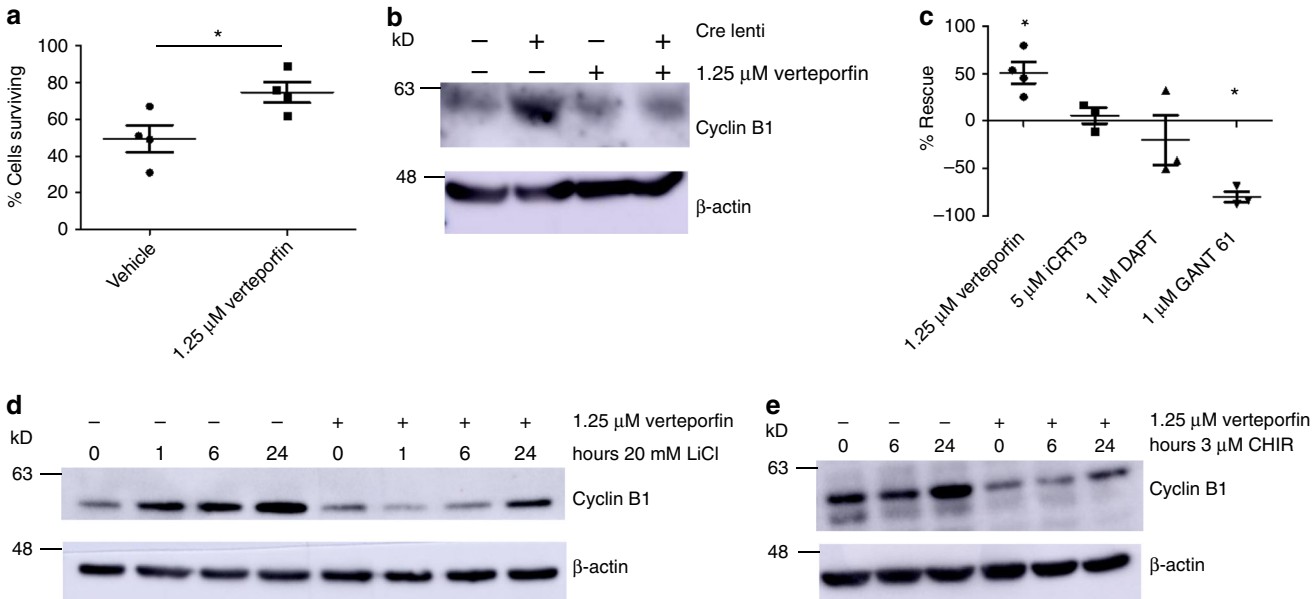

**Fig. 8** Verteporfin attenuates the effects of GSK3 loss in cultured podocytes. **a** Inhibiting YAP/TAZ activity in the nuclei of cipodGSK3DKO cells with verteporfin improves cell survival. $n = 4$ independent experiments, unpaired two-tailed $t$ test, $*p < 0.05$. See also Supplementary Fig. 10a. **b** Representative western blot showing that increased expression of Cyclin B1 in ciGSK3DKO cells is reduced by verteporfin, $n = 3$ experiments. See also Supplementary Fig. 10b. **c** Comparison of the effect of signalling pathway inhibitors on cipodGSK3DKO cell survival compared with vehicle. Inhibition of Wnt signalling with iCRT3 and Notch signalling with DAPT have no effect on cell survival while inhibition of Hedgehog signalling using GANT 61 significantly increases cell death (unpaired two-tailed $t$ test $*p < 0.05$, $n = 3$ experiments except verteporfin $n = 4$ experiments). **d, e** Representative western blot of wild-type mouse podocytes incubated for 24 h with 20 mM LiCl or 3 μM CHIR99201 showing that Cyclin B1 accumulation is reversed by treatment with 1.25 μM verteporfin, $n = 3$ experiments. See also Supplementary Figs. 10c and d. Data are presented as the mean ± SEM

Suite. Image analysis was performed with ImageJ; all images were contrast enhanced using the same parameters.

Histological abnormalities in mpodGSK3DKO mice were assessed by an experienced pathologist and scored as follows: 0 = absent; 1 = mild; 2 = moderate; 3 = severe.

Tissues for electron microscopy were fixed in 0.1 M sodium cocodylate, 2% glutaraldehyde, and imaged on a Technai 12 transmission electron microscope.

**Imaging nephrocytes.** The abdomen of adult flies was dissected open and eviscerated to reveal the heart and adjacent pericardial nephrocytes. Under phase optics, nephrocytes are readily identifiable by their characteristic size and anatomical location. Specific markers were generated by raising antisera to a peptide corresponding to amino acids 32–46 of dumbfounded, the *Drosophila* orthologue of human KIRREL/NEPH1. Tissues were counterstained with WGA and DRAQ5 to see nuclei. Fixed tissues were imaged using a Leica SP8 confocal microscope and micrographs taken with Leica Application Suite. Image analysis was performed with ImageJ; all images were contrast enhanced using the same parameters.

**Immunohistochemistry.** Tissues were fixed in 10% buffered neutral formalin, processed and paraffin embedded. Three-micrometre sections were deparaffinised in Histo-Clear and rehydrated through a graded alcohol series. Antigen retrieval was in 10 mM citrate buffer, pH6 for either 1 h at 65 °C (activated β-catenin) or by boiling for 5 min (pGSK3α/β, YAP/TAZ). Sections were quenched using 3% $H_2O_2$, followed by a blocking step in 1% BSA in PBS (activated β-catenin and pGSK3α/β) or tris buffered saline with 1% tween 20 (TBST) and 5% normal goat serum (Ajuba; YAP/TAZ; pYAPs127) for 30–45 min. Sections were incubated with primary antibodies overnight at 4 °C (activated β-catenin 1:500; pGSK3α/β 1:20; Ajuba 1:100; YAP/TAZ 1:100; pYAPs127 1:100). Sections were washed then incubated with Signalstain Boost detection reagent (Cell Signalling Technology) for 30 min at room temperature. Signalstain DAB substrate kit (Cell Signalling Technology) was applied for 1–2 min and the sections dehydrated and mounted in DPX (Sigma). Tissues were imaged using a Leica DM2000 microscope and micrographs taken with Leica Application Suite. Image analysis was performed with ImageJ; all images were contrast enhanced using the same parameters.

**IF staining of kidney sections.** Frozen kidneys were sectioned at 5 μm. Sections were blocked in phosphate buffered saline (PBS) containing 3% Bovine Serum Albumin (BSA) and 0.3% triton X-100 for 1 h, then incubated with primary antibodies overnight at 4 °C (nephrin 1:300; total β-catenin 1:100; c-myc 1:100).

Following 3 phosphate buffered saline (PBS) rinses, sections were incubated with fluorophore-conjugated secondary antibodies (Fisher) for 1 h at room temperature. Tissues were imaged using a Leica DM2000 microscope and micrographs taken with Leica Application Suite. Image analysis was performed with ImageJ; all images were contrast enhanced using the same parameters.

**Stimulation of cultured podocytes with LiCl and CHIR99201.** Conditionally immortalised mouse and human podocytes were incubated with LiCl (Sigma) at 20 mM or CHIR99201 (Sigma) at 3 μM for the times indicated.

**Western blotting.** Cultured cells or tissue samples were lysed in radio-immunoprecipitation assay (RIPA) buffer supplemented with protease and phosphatase inhibitors (Sigma). Ten to 30 μg of protein was resolved by electrophoresis then transferred to a polyvinylidene difluoride (PVDF) membrane (Millipore). Membranes were blocked in TRIS-buffered saline with 0.1% tween 20 and 5% BSA for 1 h then incubated overnight with primary antibody at a dilution of 1:1000. Membranes were washed before incubation with horseradish peroxidase conjugated secondary antibody (Sigma). Immunoreactive bands were visualised using Clarity ECL Western blotting substrate (Biorad) on a GE AI600 imager. Densitometry was performed using ImageJ software. All uncropped blots can be found in Supplementary Figs. 11–13.

**Lentiviral transduction of GSK3αβ floxed podocyte cell line.** Kidneys were isolated from GSK3αfl/flβfl/fl mice and used to make a temperature-sensitive SV40 conditionally immortalised podocyte cell line as described previously[65]. Cells were cultured at 33 °C and when 50% confluent were transduced with a lentivirus expressing Cre recombinase. Transduction was in RPMI media with hexadimethrine bromide (Sigma) at 4 μg/ml and the virus used at a multiplicity of infection of 1. Following a 24-h incubation, the lentivirus was removed and replaced with fresh media. Cells were thermo-switched to 37 °C and incubated for a further 3–7 days before imaging and protein extraction.

**Proteomics.** GSK3α/β floxed podocytes were incubated at 37 °C for 3 days following overnight transduction with Cre expressing lentivirus. Untransduced cells and wild-type mouse podocytes transduced with the lentivirus served as controls. Cells were lysed in RIPA buffer and subjected to LC-MS/MS using isobaric TMT labelling using methodology as previously described[66].

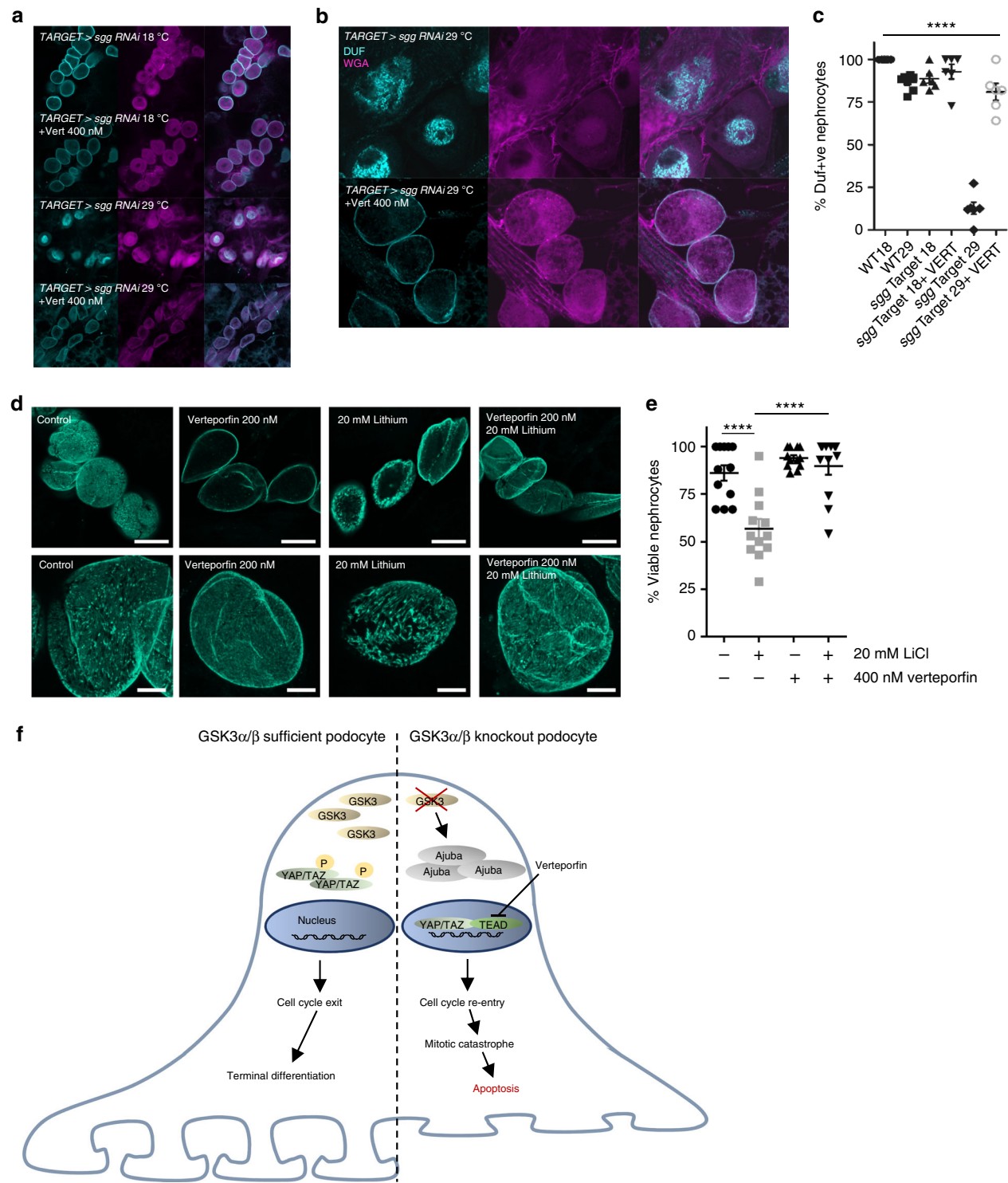

**Fixed and live imaging of cipodGSK3DKO podocytes**. The signalling pathway agonists/inhibitors Verteporfin (Tocris); iCRT3 (Sigma); DAPT (Merck Millipore) and GANT61(R and D Systems) were applied at the concentrations indicated during a 7-day incubation following 24-hour transduction of GSK3α/β floxed cells with Cre recombinase expressing lentivirus. Cells were stained with Hoechst (Sigma) at 1 µg/ml, imaged, using an IN Cell analyser 2200 (GE Healthcare) and cell number determined using IN Cell analyser workstation software (GE Healthcare).

Live cell imaging was carried out 5–7 days after viral transduction using an IncuCyte ZOOM cell imaging system (Essen Biosciences).

**YAP/TAZ nuclear translocation assay**. ciGSK3DKO podocytes or wild-type podocytes treated with LiCl or CHIR99201 were cultured in 96 well plates

(Corning). Cells were washed three times in PBS, blocked using 3% BSA, 0.3% triton X-100 for 1 h before overnight incubation at 4 °C with a YAP/TAZ antibody at 1:100. Following 3 PBS washes, cells were incubated for 1 h with fluorophore-conjugated secondary antibody. Nuclei were visualised using Hoescht at 1 µg/ml and cells imaged using an IN Cell analyser 2200 (GE Healthcare). Nuclear/cyto-plasmic fluorescence intensity was determined using In Cell analyser workstation software (GE Healthcare).

**Antibodies**. Antibodies were obtained from Cell Signaling Technology (pGSK3α (Ser21) #9316; pGSK3β(Ser9) #9323; pGSK3α/β(Ser21/9) #9331; p27Kip1 #2552; total β-catenin #8480; Cyclin B1 #12231; pcdk1(Tyr15)#4539; phosphoHistone3 (ser10)#3377; cleaved Caspase 3 #9664; PARP #9542; YAP/TAZ #8418; pYAP (ser127) #4911; Ajuba #4897), Life Technologies (total GSK3 clone 21 A); Millipore

**Fig. 9** Verteporfin attenuates the effects of GSK3 loss in *Drosophila*. **a** Knock down of *shaggy* (*sgg*) in nephrocytes prevents Duf translocation to the cell surface and is rescued by verteporfin. Flies were raised at the non-permissive temperature (18 °C, which does not allow *sgg* RNAi expression) and then cultured for 48 h at either the same temperature, or at the higher temperature (29 °C, which allows *sgg* RNAi expression). Cells were maintained in the presence or absence of 400 nM verteporfin. Nephrocytes were stained with antisera raised to the slit diaphragm protein Dumbfounded (Duf, cyan) as well as wheatgerm agglutinin (magenta). Duf was expressed at the surface of nephrocytes when RNAi was prevented but surface expression was completely prevented in when *sgg* RNAi was initiated. The lack of surface expression was rescued by the presence of verteporfin. Scale bar = 50 μm. **b** High magnification images showing rescue of surface expression by verteporfin Sale bar = 40 μm. **c** Knock down of *shaggy* (*sgg*) in nephrocytes prevents Duf translocation to the cell surface and is rescued by verteporfin. $n = 6-7$ flies per genotype. ANOVA, $**p < 0.01$ for *sgg* TARGET at 29 °C having fewer Duf + ve nephrocytes than other groups. **d** Verteporfin rescues lithium induced toxicity in nephrocytes. Adult wild-type flies were reared to between 3–7 days of age, dissected to reveal the heart and nephrocytes and cultured for 24 h in the presence or absence of lithium (20 mM), verteporfin (400 nM) or both. Scale bar top panel = 30 μm; bottom panel = 10 μm. **e** Nephrocyte viability was assessed with the vital stain calcein-AM, with nephrocytes being visualised using wheatgerm agglutinin. Lithium had a significant impact on nephrocyte viability ($****p < 0.0001$) and this was rescued by the addition of verteporfin ($****p < 0.0001$). Means compared by one-way ANOVA followed by Tukey's HSD. $n = 12$ flies from two independent experiments. **f** Cartoon showing the effect of podocyte GSK3 loss on hippo signalling. GSK3 deletion increases the expression of Ajuba, preventing the phosphorylation of YAP/TAZ and allowing its translocation into the nucleus. This results in cell cycle re-entry, mitotic catastrophe and ultimately apoptosis. Data are presented as the mean ± SEM

---

(activated β-catenin clone 8E7), Sigma (β-actin; Podocin #0372; GAPDH), Acris (nephrin), Santa Cruz Biotechnology (Synaptopodin H-140; p27kip1), and Genetex (SV40 T antigen).

**Statistics**. Statistical analysis was performed using GraphPad Prism software. When comparing two groups *t*-tests were used. When comparing more than two groups ANOVA was used with appropriate post hoc analysis. Statistical tests used, and n numbers are shown in figure legends. Data are presented as the mean and error bars represent standard error of the mean. For survival characteristics Kaplan–Meier survival plots were generated and log-rank analysis performed. *P* values less than 0.05 were deemed statistically significant.

### Data availability

The authors declare that all data supporting the findings of this study are available within the article and its supplementary information files or from the corresponding author upon reasonable request.The mass spectrometry proteomics data have been deposited in the PRIDE repository as part of the ProteomeXchange Consortium[67] under the dataset identifier PXD012145.

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

## Acknowledgements

This work was funded by Kidney Research UK, the Medical Research Council, which funds RJMC with a Senior Research Fellowship (MR/K010492/1), European Renal Association- European Dialysis Transplantation Association (ERA-EDTA), Canadian Institutes of Health Research and the National Institute of Health. Latterly it was supported by funding from the Innovative Medicines Initiative 2 Joint Undertaking under grant agreement No 115974. This Joint Undertaking receives support from the European Union's Horizon 2020 research and innovation programme and EFPIA with JDRF.

## Author contributions

The study was conceived by R.J.C., S.E.Q. and J.R.W. with expert input from P.W.M., M.A.S., G.I.W., R.J.W., P.H., J.A.H. and S.P. The in vitro and in vivo mouse experiments were performed by J.A.H., P.H., A.C.L., S.S., L.N., E.M., A.F.B., V.P. and M.M. Lithium Rat studies and the study of human kidney biopsies of patients taking lithium was performed by J.J.B. and J.P.L. under the supervision of R.J.W. The immortalised GSK3 knockout cell line was made by L.N., L.F.W. and J.U. The *Drosophila* studies were conceived and performed by P.H., viability quantification being performed by Ms Charlotte Clarke. Expert histopathology support was given by C.L.S. and A.M. The paper was written by J.A.H. and R.J.C. and then all authors read and intellectually commented on the paper. We thank Professors Jeremy Tavare and Julian Hamilton-Shields for their helpful opinions on this manuscript.

## Additional information

**Competing interests:** The authors declare no competing interests.

