## [Peer Review File · Nature Communications]

Reviewers' Comments:

Reviewer #1:

Remarks to the Author:

The paper is a sound genetic study into the requirement of both GSK3 isoforms in podocytes. Lack of both GSK3 isoforms produce kidney pathologies.

One issue to take into consideration on GSK3 research is the known existence of two pools of GSK3. One pool forms part of Wnt signaling complex and a second pool can respond to phosphorylation of Ser21/9 in GSK3a and b respectively.

While the knock-out of GSK3 isoforms do not provide information on the pathway involved, the authors have independently verified that the depletion of b-catenin (that is normally enhanced upon inhibition of GSK3b via Wnt activation) on GSK3a/b KO, does not ameliorate the condition. The overall results by the authors and previous work indeed indicate that the pathway under investigation in kidney is not mediated by Gsk3-associated to Wnt-signaling. In contrast, the authors find that the phenotype is partially recovered with inhibitors of the Hippo pathway.

The authors discuss the previous finding that long-term or high levels of Lithium in humans are linked glomerular disease and kidney failure. On the detrimental effects of GSK3 inhibition on glomerular disease the authors cite the study of lithium, which is not a specific inhibitor of GSK3. In a final sentence, the authors indicate that future use of GSK3 inhibitors should ensure adequate GSK3 function in podocytes in order to protect against severe kidney disease.

Concern:

The authors raise an important point on the potential side effects of pharmacological inhibitors of GSK3. However, those statements are not well supported because the authors have employed genetic knock-out methods but not pharmacological treatments. Did the authors test GSK3 inhibitors in their models under study?

The knock out of a protein that forms multiprotein complexes may produce different effects as compared with specific inhibitors that may not affect the formation of multiprotein complexes.

To keep the tone of the discussion on this important matter, the authors should provide results with pharmacological GSK3 inhibitors (with good/known selectivity profile) in their model systems.

Reviewer #2:

Remarks to the Author:

In this work the authors test the function of Gsk3 in the podocyte by generating a podocin-cre;Gsk3a/3b strain. Interestingly, inactivation of both copies of Gsk3a and Gsk3b is required to provide a phenotype, suggesting they might act redundantly. Mice develop proteinuria in the first week of life and podocyte foot processes are lost. Using an inducible mutant they are able to show that proteinuria also is an outcome of Gsk3a/3b inactivation in older animals, indicating that it is essential for podocyte homeostasis. An immortalized podocyte cell line is generated from the Gsk3a/3b conditional mutant and used to determine that GSK activity is required for cell viability and appears to act as a suppressor of proliferation. Proteomics screening of wild type versus mutant cells identifies the Hippo/Yap/Taz modifier Ajuba as a differentially expressed factor and the verteporfin small molecule modifier of the Yap/Taz pathway is used to partially recover the phenotype in the immortalized cells.

With caveats for the technical reservations listed, I think this is a very interesting phenotype and mechanism. On the basis of the criticism below, I do not think the authors have excluded b-catenin signaling, but the finding of Ajuba using the novel cell line is intriguing. The cell line is a novel research tool that has been used to try to explain the phenotype. For this approach to be convincing, the authors need to provide some better proof that the Ajuba protein is associated with the phenotype - for example some more convincing immunostaining data of tissue including Yap/Taz targets, proof from primary podocytes or in vivo experiments using genetics or small molecules to manipulate the Hippo pathway. Currently, the model is intriguing but not very convincing since it rests on an experiment in a single immortalized cell line. The drosophila data is interesting but it seems odd that the authors do not develop it further; because this is a more tractable genetic system it would be the most convenient with which to ask questions in vivo about the importance of the proposed Gsk3 - Hippo connection for homeostasis of the nephrocyte.

Technical points

The immunostaining for phosphorylated GSK3 in figure 1 is interesting but it lacks specificity controls.

The genetic recombination is demonstrated by a rather weak band on an ethidium bromide genotyping gel. So the efficiency of recombination is not clear.

To address both of these issues, kidney tissue from animals with single and compound mutations should be immunostained.

The use of primitive in evolutionary terms should probably be avoided. One could easily argue that drosophila are better adapted to their environment than we are, and therefore more advanced.

Most of the observations from cell culture studies are not backed up by tissue staining - for example increased proliferation and "mitotic catastrophe", Yap/Taz changes etc. There is overall a lack of marker analysis which makes this work difficult to place in the context of other published pathways.

The compound inactivation of Gsk3a/3b/b-catenin is an interesting experiment but there is no data provided demonstrating the frequency of recombination. If the frequency is low, then perhaps Gsk3a, 3b and b-catenin are being inactivated only in partially overlapping cell populations. Demonstrating loss of Gsk3a, Gsk3b, b-catenin in tissue by immunostaining or in situ hybridization is important.

Reviewer #3:

Remarks to the Author:

Hurcombe et al analyze the role of glycogen synthase kinase 3(GSK3) in podocytes, both in development and maturity. Deletion of all four GSK isoforms (both alpha and beta), either globally or in an inducible podocyte specific fashion, result in severe proteinuria with renal failure. Authors speculate that mechanistically this is caused by cell cycle re-entry mediated by dysregulation of hippo signaling.

All presented studies are technically excellent, and findings convincing.

Major Concerns:

My major concerns with the work is that it does not establish clear clinical context or significance. This significantly decreases the novelty and importance of their findings.

1) GSK3 is a major kinase with multiple known downstream targets, many of which are central downstream signaling targets. That being said, it is not entirely surprising that its complete knockout is devastating to podocyte function.

2) On the contrary, prior work from Zhou et al (JASN, 2016 Aug;27(8):2289-308) and others convincingly show that podocyte-specific deletion of both copies of GSK3 beta (GSK alpha still intact) strongly PROTECTS podocytes from injury. Likewise GSK3 inhibitors, which probably incompletely decrease GSK3 function, have also been shown to PROTECT podocytes. This dichotomy that GSK3 partial loss of function is protective, but complete loss of function catastrophic, should be addressed. In the setting of widespread testing of GSK3 inhibitors for a multitude of high profile diseases such as Alzheimer's disease, cancer, diabetes, and psychiatric disease it seems that analyzing the effect of GSK3 partial loss of function on podocytes is a more clinically relevant question than the complete loss of GSK3, as is presented in this work.

i) Have there been reports of proteinuria in patients that are treated with GSK3 inhibitors? Authors suggest in the discussion that the detrimental effects of lithium on kidney may be mediated by near complete inhibition of podocyte GSK3. Perhaps this could be tested for by immunostaining of human kidney biopsies of patients who have been on lithium long-term. It would also be interesting to test this hypothesis experimentally in lithium-treated mice.

ii) Will administration of GSK inhibitors worsen proteinuria in mouse models of podocyte injury (adriamycin, streptozocin, ect)? Prior published data suggests in fact that these drugs are podocyte protective.

Perhaps it would be worth investigating whether mice with three of four alleles of GSK deleted, are

more susceptible to proteinuria in podocyte injury models and whether there are differing effects on podocyte protection between the alpha and beta isoforms. This may help to establish the threshold levels of GSK required to sustain podocyte function.

iv) Are there any known effects of GSK expression levels in glomerular disease pathogenesis?

Minor points:

1) Authors corroborate the essential role of GSK3 both in mouse podocytes and *Drosophila* nephrocytes. The *Drosophila* data, while certainly confirmatory, is overall not surprising, and it does not move the story forward mechanistically. The main advantage of working with this organism is the relative ease and rapidity of genetic manipulation. Perhaps, the authors can use *Drosophila* to test their mechanistic hypothesis that the effects of complete GSK3 loss are mediated by diminished hippo signaling. For example, inducing combined loss of GSK and Ajuba in adult nephrocytes might mitigate the phenotype.

Reviewers' comments:

Referee #1

The paper is a sound genetic study into the requirement of both GSK3 isoforms in podocytes. Lack of both GSK3 isoforms produce kidney pathologies.

One issue to take into consideration on GSK3 research is the known existence of two pools of GSK3. One pool forms part of Wnt signaling complex and a second pool can respond to phosphorylation of Ser21/9 in GSK3a and b respectively. While the knock-out of GSK3 isoforms do not provide information on the pathway involved, the authors have independently verified that the depletion of b-catenin (that is normally enhanced upon inhibition of GSK3b via Wnt activation) on GSK3a/b KO, does not ameliorate the condition. The overall results by the authors and previous work indeed indicate that the pathway under investigation in kidney is not mediated by Gsk3-associated to Wnt-signaling. In contrast, the authors find that the phenotype is partially recovered with inhibitors of the Hippo pathway.

The authors discuss the previous finding that long-term or high levels of Lithium in humans are linked glomerular disease and kidney failure. On the detrimental effects of GSK3 inhibition on glomerular disease the authors cite the study of lithium, which is not a specific inhibitor of GSK3. In a final sentence, the authors indicate that future use of GSK3 inhibitors should ensure adequate GSK3 function in podocytes in order to protect against severe kidney disease.

Concern:

The authors raise an important point on the potential side effects of pharmacological inhibitors of GSK3. However, those statements are not well supported because the authors have employed genetic knock-out methods but not pharmacological treatments. Did the authors test GSK3 inhibitors in their models under study? The knock out of a protein that forms multiprotein complexes may produce different effects as compared with specific inhibitors that may not affect the formation of multiprotein complexes.

To keep the tone of the discussion on this important matter, the authors should provide results with pharmacological GSK3 inhibitors (with good/known selectivity profile) in their model systems. On the detrimental effects of GSK3 inhibition on glomerular disease the authors cite the study of lithium, which is not a specific inhibitor of GSK3.

RESPONSE: Thank you for this very helpful and constructive suggestion. We now include a series of studies examining the importance of pharmacological inhibition of GSK3 in both the podocyte and nephrocyte. We omitted some of this data in our initial submission as we thought it made the paper too long and cumbersome and were planning to submit this data in a new manuscript in the future, after the initial genetically focused work had been published. However, we agree that combining the pharmacological and genetic data is an excellent idea and strengthens this manuscript greatly. As the paper is now very big we

intend taking out the first figure showing that insulin phosphorylates GSK3 alpha and beta in the podocyte. We think this data is not critical for this manuscript.

To address the importance of the pharmacological inhibition of GSK3 in the podocyte/nephrocyte we now include a new results section entitled “Prolonged pharmacological inhibition of GSK3 α and β kinase activity with lithium causes a glomerulopathy in mammals and severe nephrocyte dysfunction in *Drosophila*.” We also include extra data on another pharmacological GSK3 inhibitor CHIR99201 elsewhere in the manuscript.

Specifically we include the following data:

- A. That Lithium causes inhibitory phosphorylation of GSK3 α and β in both murine and human podocytes (**NEW SUPPLEMENTARY FIGURES 4A-D**). We agree with the reviewer that lithium is a “dirty” GSK3 inhibitor but clinically it is very important as many patients receive this medication for bi-polar conditions, hence our rationale for studying it.
- B. That chronic exposure of lithium in rats causes inhibitory phosphorylation of GSK3 $\alpha\beta$ together with activation of β -catenin in the podocytes of these animals (**NEW FIGURE 3A**). These rats also develop significant proteinuria and evidence of glomerulosclerosis (**NEW FIGURES 3B, 3C AND 3D**). This is similar as reported in human patients chronically treated with lithium (Markowitz, G.S., *et al.* Lithium nephrotoxicity: a progressive combined glomerular and tubulointerstitial nephropathy. *J Am Soc Nephrol* **11**, 1439-1448 (2000). Aurell, M., *et al.* Renal function and biopsy findings in patients on long-term lithium treatment. *Kidney Int* **20**, 663-670 (1981).) This work was performed in New Zealand by Dr Jenny Bedford and Dr John Leader under the supervision of Professor Robert Walker. Hence, they are now included as co-authors.
- C. That *drosophila* nephrocytes exposed to lithium causes an alteration in the shape, number and function of these cells in a dose dependent manner (**NEW Figures 3E, 3F, 3G, 3H, and 3I**).
- D. A biopsy from a patient on long-term lithium who had renal compromise showed inhibitory phosphorylation of GSK3 $\alpha\beta$ (**NEW FIGURE 3J**)
- E. That *in vitro* pharmacological inhibition of GSK3 $\alpha\beta$ in the podocyte (using either Lithium or the more specific GSK3 inhibitor CHIR99201) alters hippo signalling by causing Ajuba to be up-regulated (**NEW FIGURE 6C; 6D**), YAP/TAZ to translocate into the nuclei of the cells (**NEW FIGURES 6F AND 6G**), and Cyclin B1 to accumulate (**NEW FIGURES 6C; 6D**). These effects were exactly the same as in the genetically knocked down GSK3 $\alpha\beta$ cells. Furthermore, we functionally rescued the phenotypes we observe in these pharmacological models by co administrating Verteporfin to these systems (**NEW FIGURES 8D AND 8E**) and also in lithium treated nephrocytes (**NEW FIGURES 8I and 8J**). This strengthens the link between GSK3 $\alpha\beta$ inhibition in the podocyte/nephrocyte and the role of the hippo pathway.

We think including the pharmacological data greatly strengthens this manuscript, so thank the reviewer for suggesting this.

Referee #2

In this work the authors test the function of GSK3 in the podocyte by generating a podocin-cre;Gsk3a/3b strain. Interestingly, inactivation of both copies of Gsk3a and Gsk3b is required to provide a phenotype, suggesting they might act redundantly. Mice develop proteinuria in the first week of life and podocyte foot processes are lost. Using an inducible mutant they are able to show that proteinuria also is an outcome of Gsk3a/3b inactivation in older animals, indicating that it is essential for podocyte homeostasis. An immortalized podocyte cell line is generated from the Gsk3a/3b conditional mutant and used to determine that GSK activity is required for cell viability and appears to act as a suppressor of proliferation. Proteomics screening of wild type versus mutant cells identifies the Hippo/Yap/Taz modifier Ajuba as a differentially expressed factor and the verteporfin small molecule modifier of the Yap/Taz pathway is used to partially recover the phenotype in the immortalized cells.

With caveats for the technical reservations listed, I think this is a very interesting phenotype and mechanism. On the basis of the criticism below, I do not think the authors have excluded b-catenin signaling, but the finding of Ajuba using the novel cell line is intriguing. The cell line is a novel research tool that has been used to try to explain the phenotype.

Major Critique 1. *For this approach to be convincing, the authors need to provide some better proof that the Ajuba protein is associated with the phenotype - for example some more convincing immunostaining data of tissue including Yap/Taz targets, proof from primary podocytes or in vivo experiments using genetics or small molecules to manipulate the Hippo pathway.*

RESPONSE: Many thanks for this helpful suggestion. We have addressed this at multiple levels:

A. Better proof that Ajuba is increased with the inhibition of GSK3 in the podocyte.

We now show that:

Ex-vivo

- Ajuba increases in the podocytes of both the developmental and mature GSK3 $\alpha\beta$ knockout models using both immunohistochemistry (**NEW FIGURE 7C; 7D**) and also western blotting of glomerular lysates (**NEW FIGURE 7A; 7B**).
- Pharmacologically inhibiting GSK3 in the rat and in human patients with lithium causes podocyte Ajuba to increase (**New FIGURE 7G**).

in-vitro

- In addition to genetically inhibiting GSK3 $\alpha\beta$ (**FIGURE 6B**) in the podocyte pharmacological inhibition with either lithium or CHIR99201 causes Ajuba to be up regulated here (**NEW FIGURES 6C; 6D**).

- B. ***The key event that occurs when hippo signalling is inhibited (as occurs when Ajuba increases) is the nuclear translocation of YAP/TAZ.*** We now include immunohistochemical data that this occurs in both the developmental (**FIGURE 7C**) and mature mouse genetic GSK3 $\alpha\beta$ knockout models (**FIGURE 7D**). This also occurs *in vitro* in the genetic GSK3 $\alpha\beta$ knockdown podocyte cell lines (**FIGURE 6E**) as well as lithium and CHIR99201 treated podocyte cell lines (**NEW FIGURES 6F AND 6G**).
- C. ***Regarding YAP/TAZ targets*** we show that Cyclin A2 and c-myc are upregulated in all of the models (Chen et al. Oncotarget. 2018 Jan 2; 9(1): 668–679; Zanconato et al. *Nat Cell Biol* 2015 **17**, 1218-1227.) (**NEW FIGURES 7E and 7F**). We think that detection of many of the classic hippo targets is difficult as the cells try and enter the cell cycle and then die making them impossible to detect.

Major Critique 2. Currently, the model is intriguing but not very convincing since it rests on an experiment in a single immortalized cell line. The drosophila data is interesting but it seems odd that the authors do not develop it further; because this is a more tractable genetic system it would be the most convenient with which to ask questions in vivo about the importance of the proposed Gsk3 - Hippo connection for homeostasis of the nephrocyte.

RESPONSE: This is another excellent point and suggestion. We have now studied the inducible genetic shaggy (GSK3) knockdown drosophila nephrocytes for the effects of inhibiting the YAP/TAZ – TEAD axis here with Verteporfin. When we incubate the shaggy knockdown nephrocytes with Verteporfin they are completely protected from mis-localising Dumfounded and maintain their structure (**NEW FIGURE 8F**). Furthermore, verteporfin protects nephrocytes exposed to the pharmacological GSK3 inhibitor lithium in their diet (**New FIGURE 8G**).

This is further functional evidence that the hippo pathway is linked to GSK3 activity in nephrocyte cells. It supplements our *in vitro* podocyte genetic GSK3 alpha/beta knockdown data very nicely.

Technical points

1. The immunostaining for phosphorylated GSK3 in figure 1 is interesting but it lacks specificity controls.

RESPONSE: We are now going to leave this figure out. However, it is an excellent point. We excluded isotype matched antibody controls in the initial submission which we have done.

We have added in more data to show that we have knocked out GSK3 alpha and beta using immunohistochemistry instead of this figure. This is included as **NEW SUPPLEMENTARY FIGURE 1C**

2. The genetic recombination is demonstrated by a rather weak band on an ethidium bromide genotyping gel. So the efficiency of recombination is not clear.

RESPONSE: We have made this clearer. (**SUPPLEMENTARY FIGURE 1B**).

3. To address both of these issues, kidney tissue from animals with single and compound mutations should be immunostained.

RESPONSE: We have done this using immunohistochemistry. (**NEW SUPPLEMENTARY FIGURE 1C**)

4. The use of primitive in evolutionary terms should probably be avoided. One could easily argue that drosophila are better adapted to their environment than we are, and therefore more advanced.

RESPONSE: This is a good point. We have removed “more primitive” in respect to drosophila throughout the manuscript.

5. Most of the observations from cell culture studies are not backed up by tissue staining - for example increased proliferation and “mitotic catastrophe”, Yap/Taz changes etc. There is overall a lack of marker analysis which makes this work difficult to place in the context of other published pathways.

RESPONSE: Thank you for these points. We have addressed the YAP/TAZ changes in the animal models and also the genetic and pharmacologically manipulated cell lines (See response to major critique 1 B above). We are sure the cells are entering the cell cycle as evidenced by increased binuclear podocytes detected *in vitro* (FIGURE 5D) and increased DNA production using PCNA IHC (**New FIGURE 5G**) in both our developmental and mature GSK3 knockout models. We also include a new picture *in vivo* of a mitotic figure in a mature GSK3 knockout mouse (**NEW FIGURE 5H**) but this is much more challenging to detect in comparison to the *in vitro* models as it is probably a rare occurrence and once it happens the cell dies and the signal is lost.

6. The compound inactivation of Gsk3a/3b/b-catenin is an interesting experiment but there is no data provided demonstrating the frequency of recombination. If the frequency is low, then perhaps Gsk3a, 3b and b-catenin are being inactivated only in partially overlapping cell populations. Demonstrating loss of Gsk3a, Gsk3b, b-catenin in tissue by immunostaining or in situ hybridization is important.

RESPONSE: Thank you for this point. We apologise for not making this data clearer in our initial submission as using PCR, IHC and IF (**Supplementary FIGURES 5 c, d and e**) we were able to show that beta catenin was knocked down in our triple GSK3 α / β / β - catenin model. We have now added this information to the main text in the results (section “Combined GSK3 α and β loss in the podocyte activates Wnt- β -catenin but this is not responsible for pathology”) to make it easier to appreciate for the reader.

The fact that both the developmental and mature triple knockout models were not rescued from kidney disease when β – catenin was knocked down suggests that GSK3 α and β were both adequately knocked down. We agree if the model had been rescued it would have been very important to show that there were similar levels of GSK3 alpha and/or beta expression in the triple compared to the double GSK3 isoform knockout systems, as the increased GSK3 expression may have explained the better outcome, but this was not the case.

Reviewer #3 (Remarks to the Author):

Hurcombe et al analyze the role of glycogen synthase kinase 3(GSK3) in podocytes, both in development and maturity. Deletion of all four GSK isoforms (both alpha and beta), either globally or in an inducible podocyte specific fashion, result in severe proteinuria with renal failure. Authors speculate that mechanistically this is caused by cell cycle re-entry mediated by dysregulation of hippo signaling.

All presented studies are technically excellent, and findings convincing.

RESPONSE: Thank you for these complimentary comments.

Major Concerns:

My major concern with the work is that it does not establish clear clinical context or significance. This significantly decreases the novelty and importance of their findings.

RESPONSE: This is a fair point but we think that by adding in our pharmacological data it now makes this work much more clinically relevant. That stated we think it also important to emphasise that this paper is not a purely “clinical” paper but also studying fundamental podocyte cell biology in respect to GSK3 manipulation which we believe is also strong and novel.

1) GSK3 is a major kinase with multiple known downstream targets, many of which are central downstream signaling targets. That being said, it is not entirely surprising that its complete knockout is devastating to podocyte function.

RESPONSE: Again, a fair point but as is highlighted in point 2 (below) by the reviewer there is a lot of recent published data suggesting that inhibiting GSK3 in the podocyte may be beneficial for renal function. This is in context of a major drive to develop new GSK3 inhibitors for conditions such as cancer, diabetes and Alzheimer’s disease.

Our work clearly shows that excessive genetic and pharmacological inhibition of both GSK3 isoforms in the podocyte/nephrocyte is highly detrimental. Going forward it will be important to establish which pathways are beneficial and which are detrimental downstream of each GSK3 isoform as this may reveal new therapeutic targets for kidney diseases. Regardless of this, our work shows GSK3 is crucial for the health of podocyte, in preventing it from entering the cell cycle and also links GSK3s action with hippo signalling which is all conceptually novel.

2) On the contrary, prior work from Zhou et al (JASN, 2016 Aug;27(8):2289-308) and others convincingly show that podocyte-specific deletion of both copies of GSK3 beta (GSK alpha still intact) strongly PROTECTS podocytes from injury. Likewise GSK3 inhibitors, which probably incompletely decrease GSK3 function, have also been shown to PROTECT podocytes. This dichotomy that GSK3 partial loss of function is protective, but complete loss of function catastrophic, should be addressed. In the setting of widespread testing of GSK3 inhibitors for a multitude of high profile diseases such as Alzheimer's disease, cancer, diabetes, and psychiatric disease it seems that analyzing the effect of GSK3 partial loss of function on podocytes is a more clinically relevant question than the complete loss of GSK3, as is presented in this work.

RESPONSE: Thank you we agree that this nice work has suggested that partial inhibition of podocyte GSK3 function may be a good therapeutic strategy for kidney disease. However, these papers have not addressed the role of GSK3 α in great detail (indeed some of the papers have suggested it is not present in the podocyte). We think our work is important to show that excessive inhibition of GSK3 α and β in the podocyte is highly detrimental (both genetic and pharmacological). Importantly there are no isoform specific GSK3 inhibitors with all current inhibitors suppressing both isoforms to a greater or lesser extent. The nice work of Zhou which is currently in the literature proposes that pharmacological GSK3 inhibitors are protective against renal disease and a likely possibility is that with this work clinicians / scientists may think that by increasing the dose of pharmacological inhibition of GSK3 it may be beneficial in treating kidney disease (s). However, our work suggests this could be highly detrimental. This is an important message.

We agree that going forward understanding the specific downstream signalling pathways elicited by the different isoforms will be potentially highly informative. However this is another set of intense experiments, which will be a natural extension of this work. We have articulated this in the discussion.

3) Have there been reports of proteinuria in patients that are treated with GSK3 inhibitors?

RESPONSE: Yes, there are several reports showing not only proteinuria but Focal Segmental glomerulosclerosis (FSGS) occur in patients on long-term lithium therapy (Markowitz, G.S., et al. Lithium nephrotoxicity: a progressive combined glomerular and tubulointerstitial nephropathy. *J Am Soc Nephrol* **11**, 1439-1448 (2000) Hansen, H.E., et al. Renal function and renal pathology in patients with lithium-induced impairment of renal concentrating ability. *Proc Eur Dial Transplant Assoc* **14**, 518-527 (1977). Aurell, M., et al. Renal function and biopsy findings in patients on long-term lithium treatment. *Kidney Int* **20**, 663-670 (1981)).

4) Authors suggest in the discussion that the detrimental effects of lithium on kidney may be mediated by near complete inhibition of podocyte GSK3. Perhaps this could be tested for by immunostaining of human kidney biopsies of patients who have been on lithium long-term. It would also be interesting to test this hypothesis experimentally in lithium-treated mice.

RESPONSE: Thank you for these suggestions. We have now performed immunohistochemistry on a kidney biopsy from a patient who was biopsied on long-term lithium and this shows evidence of GSK3 inhibitory phosphorylation (**NEW FIGURE 3J**) and also Ajuba upregulation in the podocyte (**NEW figure 7G**)

We have found exactly the same in a cohort of rats given long-term lithium therapy (Please see response to expert reviewer 1).

4) Will administration of GSK inhibitors worsen proteinuria in mouse models of podocyte injury (adriamycin, streptozocin, ect)? Prior published data suggests in fact that these drugs are podocyte protective.

Perhaps it would be worth investigating whether mice with three of four alleles of GSK deleted, are more susceptible to proteinuria in podocyte injury models and whether there are differing effects on podocyte protection between the alpha and beta isoforms. This may help to establish the threshold levels of GSK required to sustain podocyte function.

RESPONSE: This is an interesting set of experiments. We know that losing 3 out of the 4 GSK3 $\alpha\beta$ alleles is not detrimental in the non-stressed state but have not tested them in conditions of stress. We agree with the reviewer that it will probably depend on the degree of GSK3 suppression in the system. This is something that we will consider doing in the future but within the time frame of revising this paper (6 months) it is impossible to generate enough numbers of mice to robustly perform these experiments.

5) Are there any known effects of GSK expression levels in glomerular disease pathogenesis?

RESPONSE: This is another very interesting question. We have looked at nephroseq which examines the glomerular mRNA levels of different disease patient groups. There is no obvious pattern in GSK3 isoform expression here except that commonly when one of the GSK3 isoforms are suppressed at the mRNA level the other isoform seems to go up. This may be compensatory. However, assessing if functional GSK3 activity changes in the podocyte in disease situations is very difficult as it is controlled by both its kinase activity and scaffolding properties. Therefore, there may be no differences in the mRNA or protein levels of this enzyme, but functional differences (such as phosphorylation effects). Going forward we may be able to model the array of downstream changes caused by a loss of the isoforms and is something we will consider doing.

Minor points:

1) Authors corroborate the essential role of GSK3 both in mouse podocytes and Drosophila nephrocytes. The Drosophila data, while certainly confirmatory, is overall not surprising, and it does not move the story forward mechanistically. The main advantage of working with this organism is the relative ease and rapidity of genetic manipulation. Perhaps, the authors can use Drosophila to test their mechanistic hypothesis that the effects of complete GSK3 loss are mediated by diminished hippo signaling. For example, inducing combined loss of GSK and Ajuba in adult nephrocytes might mitigate the phenotype.

RESPONSE: Thank you for this comment which we have now addressed. Please see response to reviewer 2. Major critique 2. We have now found that inhibiting YAP/TAZ in the nephrocyte protects against the detrimental effects caused by both genetic and pharmacological inhibition of GSK3 in this cell type.

Reviewers' Comments:

Reviewer #1:

Remarks to the Author:

The authors have addressed all my comments. The addition of experiments with lithium and an alternative GSK3 inhibitor, together with other additions have substantially improved the manuscript.

Reviewer #2:

Remarks to the Author:

The authors have addressed my comments and I think this is a very intriguing study.

Reviewer #3:

Remarks to the Author:

The revised manuscript is dramatically improved. The addition of data using pharmacological GSK3 inhibition dramatically strengthens their overall findings. All other questions were adequately addressed. The work appears to be ready for publication at this point.